# Data-constrained assessment of ocean circulation changes since the middle Miocene in an Earth system model

Katherine A. Crichton[1]*, Andy Ridgwell[2], Daniel J. Lunt[3], Alex Farnsworth[3], Paul N. Pearson[1]

[1]School of Earth and Ocean Sciences, Cardiff University, UK
2Department of Earth and Planetary Sciences, University of California, Riverside, CA 92521, USA
3School of Geographical Sciences, University of Bristol, Bristol, UK
* Now at School of Geography, University of Exeter, EX4 4RJ, UK

*Correspondence to*: Katherine A. Crichton (k.a.crichton@exeter.ac.uk)

**Abstract.** Since the middle Miocene (15 Ma, million years ago), the Earth's climate has undergone a long-term cooling trend,
characterised by a reduction in ocean temperatures of up to 7 to 8 °C. The causes of this cooling are primarily thought to be
due to tectonic plate movements driving changes in large-scale ocean circulation patterns, and hence heat redistribution, in
conjunction with a drop in atmospheric greenhouse gas forcing (and attendant ice-sheet growth and feedback). In this study
we assess the potential to constrain the evolving patterns of global ocean circulation and cooling over the last 15 Ma by
assimilating a variety of marine sediment proxy data in an Earth system model. We do this by first compiling surface and
benthic ocean temperature and benthic carbon-13 ($\delta^{13}C$) data in a series of seven time-slices spaced at approximately 2.5
million year intervals. We then pair this with a corresponding series of tectonic and climate boundary condition reconstructions
in the cGENIE ('muffin' release) Earth system model, including alternative possibilities for an open vs. closed Central
American Seaway (CAS) from 10 Ma onwards. In the cGENIE model, we explore uncertainty in greenhouse gas forcing and
the magnitude of North Pacific to North Atlantic salinity flux adjustment required in the model to create an Atlantic Meridional
Overturning Circulation (AMOC) of a specific strength, via a series of 12 (one for each tectonic reconstruction) 2D parameter
ensembles. Each ensemble member is then tested against the observed global temperature and benthic $\delta^{13}C$ patterns. We
identify that a relatively high $CO_2$ equivalent forcing of 1120ppm is required at 15 Ma in cGENIE to reproduce proxy
temperature estimates in the model, noting that this $CO_2$ forcing is dependent on cGENIEs climate sensitivity and that it
incorporates the effects of all greenhouse gases. We find reproducing the observed long-term cooling trend requires a
progressively declining greenhouse gas forcing in the model. In parallel to this, the strength of the AMOC increases with time
despite a reduction in salinity of the surface North Atlantic over the cooling period, attributable to falling intensity of the
hydrological cycle and to lowering polar temperatures, both caused by $CO_2$-driven global cooling. We also find that a closed
CAS from 10 Ma to present shows better agreement between benthic $\delta^{13}C$ patterns and our particular series of model
configurations and data. A final outcome of our analysis is of a pronounced ca. 1.5 ‰ decline occurring in atmospheric (and
ca. 1 ‰ ocean surface) $\delta^{13}C$ that could be used to inform future carbon-13 based proxy reconstructions.

# 1 Introduction and background

Since the middle Miocene (~ 15 Ma), the Earth has experienced a pronounced and quasi continuous period of global cooling, characterized by an expansion of ice sheets over Antarctica and later, the establishment of the Greenland ice sheet (Zachos et al., 2008; Cramer et al., 2011). The Pleistocene (2.6 Ma to present) also saw the intensification of glacial-interglacial cycles and episodically, the establishment of the North American ice sheet. Terrestrial temperature proxies indicate that the Miocene was significantly warmer than the present day (Pound et al., 2012 and references therein). Marine data also indicate a significantly warmer-than-present Miocene climate, with global surface ocean temperatures 6 °C warmer than present (Stewart et al., 2004; Herbert et al., 2016), and global deep ocean temperature 4 to 6 °C warmer than present (Cramer et al., 2011). ~~E~~Although ~~e~~early proxy estimates of atmospheric $CO_2$ in the mid-Miocene show levels may have ~~even~~ been similar to or even ~~been~~ lower than present (Pagani et al 2005, Pearson and Palmer 2000). ~~C.~~climate modelling studies have tended to apply a forcing of around 400ppm for the general Miocene period (but not necessarily to represent peak mid-Miocene warmth) (Burls et al., 2021; Londoño et al. 2018 references therein). ~~Climate modelling studies have tended to apply a forcing of around 400ppm, based on later estimates (Londoño et al 2018 references therein), and slightly lower than a recent multi-proxy estimate of mid-Miocene $CO_2$ that places it at 450 to 550 ppm (Steinthordottir et al 2021). Other~~ More recent work places ~~m~~Mid-Miocene $CO_2$ higher still, from ~500pm to ~1000ppm although uncertainty still exists (Londoño et al 2018; Sosdian et al., 2018; Stoll et al 2019).Regardless,~~; but~~ a falling atmospheric $CO_2$ is thought to have been a driver of global cooling since the middle Miocene (Rae et al 2021). ~~Fossil Neotropical leaf assemblages suggest $CO_2$ mixing ratios of 528ppm (low mode) and 912 ppm (high mode) (Londoño et al 2018). Using the boron isotope pH proxy, $CO_2$ mixing ratios have been estimated at between 470 to 630ppm (Sosdian et al., 2018). Re-calculating the alkenone $\epsilon_P$ $CO_2$ proxy whilst taking account of upregulation of carbon concentrating mechanisms in phytoplankton results in around 1000 ppm for the mid-Miocene (Stoll et al 2019).~~

Associated with this interval of global cooling, fundamental changes have also occurred in large scale ocean circulation patterns (Butzin et al., 2011). Specifically – the Atlantic Meridional Overturning Circulation (AMOC), that today redistributes heat to the Northern hemisphere at the expense of the South, is thought to have established in its current form sometime after the middle Miocene (Sepulchre et al., 201~~4~~3; Bell et al., 2015) (but not necessarily seeing its first appearance – see: Abelson and Eres 2017). This 'switching on' of a prominent AMOC is suspected to have been linked to the closing of the seaway between the Atlantic and the Pacific with the creation of the Isthmus of Panama (Lunt et al., 2008; Montes et al., 2015; O'Dea et al., 2016; Jaramillo et al., 2017) that finally closed the Central American Seaway (CAS). Other seaways have also been tectonically transformed since the Miocene, particularly: the disappearance of the Tethys Sea due to the northward movement of Africa (Rogl et al., 1999; Hamon et al., 2013; Bialik et al., 2019), the restriction of the Indonesian seaway with the northward movement of Australia (Srinivasan et al., 1998), and the widening of the Drake Passage driven by the northward movement of South America relative to Antarctica (Lagabrielle et al., 2009). Associated with these plate movements, the Miocene to Holocene was characterised by significant mountain building which may have played a direct role in the draw-

down of atmospheric $CO_2$ via weathering and hence potentially playing a key role in the progressive cooling (Filippelli et al., 1997; Raymo et al., 1998), although this is far from certain (Steinthorsdottir et al 2020).

Putting all these changes together, Miocene climate modelling efforts have found that reconciling the combined constraints of ocean temperature, $CO_2$ indicators and Antarctic ice sheet dynamics is a non-trivial task (Micheels et al., 2009; Henrot et al., 2010, Bradshaw et al., 2012; Sijp et al., 2014). Vegetation feedbacks have also likely been integral in creating the Miocene climatic conditions (Henrot et al., 2010; Knorr et al., 2011; Micheels et al., 2011; Bradshaw et al., 2015) in addition to altered bathymetry, topography and atmospheric $CO_2$ as drivers of a warmer climate.

In this paper we aim to step back from the spatial and dynamical complexities of the land surface and terrestrial biosphere and interaction with the atmosphere, and focus on exploring to what extent proxy data can constrain changing ocean circulation and temperature patterns in conjunction with a simplified forcing of surface climate, in using an Earth system model (of 'intermediate complexity'). We do not to provide an exhaustive sensitivity study of Miocene seaways and details of the marine carbon cycle here (. Ffor an overview of Miocene circulation and the effects of seaways on that circulation see, e.g. Butzin et al. (2011) and Sijp et al. (2014), a . A full review of $\delta^{13}C$ and $\delta^{18}O$ as ocean circulation tracers used to infer seaway and global ice volume and climate changes in the Cenozoic is available in MacKensen and Schmiedl 2019, and a review of Miocene climate and data indicators in Steinthorsdottir et al. 2020). Instead, we aim to combine constraints of temperature and ocean circulation to create plausible and self-consistent paleo realisations of climate and ocean circulation, for each of seven time-slices spanning the middle Miocene to the late Holocene. Our focus in this paper is hence on the model-data methodology plus self-consistency and plausibility of the outcome.

## 2 Model-data Methodology

In summary, oOur overall methodology is to is based on creatinge a suite of data-sets that can help constrain the large-scale patterns of ocean circulation and temperature (and hence climate in a broad sense). We compile and employ foraminifera proxy data for: surface temperature, benthic $\delta^{18}O$ (to calculate benthic temperature), and benthic $\delta^{13}C$. The surface and benthic (sea-floor) temperature data allows us to evaluate the model skill in reproducing ocean heat distribution, and hence the models simulated global-scale pattern of ocean circulation. It also provides a means of determining the atmospheric forcing (in terms of $CO_2$) that produces surface and deep ocean temperatures that best match the data. We further compare modelled benthic global mean ocean temperature to a published estimate from Cramer et al (2011) that used the Mg/Ca thermometer. The isotope of carbon, carbon-13, has been used as a tracer for paleo ocean circulation for many years (Lynch-Stieglitz 2003), and we employ it here also as a further circulation constraint. Different water masses have characteristic $\delta^{13}C$ signatures, which depend on biological activity (in the ocean carbon pumps) and critically for this study, also on ocean circulation patterns. Rather than attempt a continuous model-data analysis for the entire 15 Ma long interval, we discretize the mid Miocene to the present (late Holocene) (Table 1) in a series of 7 time slices.

For convenience, uncertainty in model climate forcing is assumed to be represented by atmospheric $CO_2$ – this should to be understood to be as an 'equivalent' $CO_2$ forcing that encompasses changes in all atmospheric greenhouse gases (esp. methane). Ocean circulation in the model is a function of surface ocean boundary conditions (esp. wind stress) and climate (and the atmospheric $CO_2$ forcing), ocean bathymetry, and the existence and nature of seaways and gateways – all of which we adjust at each time-slice (along with the solar constant). We further introduce a variable (parameter uncertainty) to the

model that alters the salinity transfer between North Pacific and North Atlantic – a classic 'flux adjustment'. This represents the effect of atmospheric moisture transport and the relative precipitation that East vs. West draining watersheds receive on the north American continental land mass, none of which can be resolved well in the simple 2D energy-moisture balance model based approach employed here (Edwards and Marsh, 2005; Marsh et al., 2011). This 'flux correction' results in an Atlantic Meridional Overturning Circulation (AMOC) being induced (and/or strengthened) in cGENIE and hence represents a

parameter 'knob' in trying to find a fit to ocean proxy data. (Indeed, it is required in the standard present-day configuration (Ridgwell et al., 2007; Marsh et al., 2011) in order to rectify the aforementioned simplification in modelled atmospheric moisture transport and dynamics.)

## 2.1 Datasets

We selected published surface ocean temperature data based on either alkenones or TEX86 for all seven slices (with the exception of two datapoints at 15 Ma) (Fig. 1 shows surface temperature data locations and Supplementary material A contains the dataset), noting that our selected proxies, like any proxy, are still subject to uncertainties and limitations, such as seasonality effects, surface/subsurface waters representation, or nonthermal influences (discussed in Richey and Tierney 2016 for $U_{37}^K$ and TEX$_{86}$). (Fig. 1 shows surface temperature data locations and Supplementary material A contains the dataset. For slices

2.5 Ma to 15 Ma, published benthic $\delta^{13}C$ and $\delta^{18}O$ data were selected only for the *Cibicidoides* and *Planulina* foraminifera species. These species were selected so that temperature could be calculated from shell $\delta^{18}O$ using the linear model from Marchitto et al. (2014), to reduce uncertainty driven by species-specific effects on shell isotopic signatures. We ensured that at least 15 data-points were available for each time-slice and that coverage included the Atlantic and Pacific basins ( – Fig. 1 shows benthic data locations, Fig. 2 the $\delta^{13}C$ data plotted by paleo-latitude, and Supplementary material B for the dataset).

Final benthic temperatures calculated from $\delta^{18}O$ take account of the effect of benthic water salinity on $\delta^{18}O_{sw}$ which is in turn influenced by ocean circulation (the calculated temperatures in Table S2 are uncorrected for salinity) and we use the modelled benthic salinity field to create the correction for the local water $\delta^{18}O_{sw}$. The method used in this calculation is described in detail in Appendix A. Paleo-locations for each data point were found using the reconstructions in www.paleolocation.org (Urban and Hardisty 2013) that provide Zanclean (4.466 Ma), Tortonian (9.427 Ma) and Burdigalian (18.2 Ma) paleo-locations

using PLATES reconstructions (University of Texas). Thereafter, the paleo-locations for each data-point at our time-slices (Table 1) were interpolated using these three points together with the present day location in a cubic model regression.

To account for age model uncertainties in the isotope data, a window of ±1 million years was inspected for each $\delta^{18}O$ and $\delta^{13}C$ data-point to ensure the data value was not unrepresentative of the general value at that site around the target age (rather than taking a mean value over a certain window of time). This approach was applied to try to ameliorate the effect of uncertainties in age models, particularly for periods in which isotope values are changing significantly over the 2 million year intervals. In datasets with high resolution (i.e. showing possible orbital type climate cycling) and generally better age models, an intermediate value between the high and low nearest to the target age was selected (so targeting mid-climate states). All the data points and their paleo-locations, and the ±million year time-series plotted for each data location are available in Supplementary Material C, Fig. S1.

We created datasets for the late Holocene time-slice slightly differently. For this, we took the Holocene benthic $\delta^{13}C$ from Peterson et al. (2014). For ocean temperature, we used data from the World Ocean Atlas (WOA) 2009 (Levitus et al. 2010), assuming that deep ocean temperature has changed little between the late Holocene and the present. Datapoints for the deep ocean (at 4000m) were extracted from WOA 2009 to create the benthic temperature dataset for model-data fit statistics (only data locations where cGENIEs seafloor depth closely matched the WOA 2009 seafloor depth were used).

## 2.2 The cGENIE.muffin Earth system model

We run the open source intermediate complexity Earth system model cGENIE ('muffin' release) (see Code Availability section). As implemented in our study, this comprises: (1) a 3D ocean circulation model configured on a 36×36 equal area grid, with 16 non-equally spaced vertical levels in the ocean, (2) a 2D energy-moisture balance atmospheric model ('EMBM') component, and (3) a 2D dynamic-thermodynamic sea-ice model component. These three individual components and their coupling are described in Marsh et al. (2011) (and references therein). The basic physics parameter calibration of the climate model component is as per Cao et al. (2009) unless described otherwise. For the modern Atlantic basin configuration (and, as per the findings of this paper, also over the past 15 Ma), the lack of both atmospheric dynamics and explicit topography on land in our configuration model requires that a freshwater flux adjustment is applied. This is implemented in the modern configuration of the model, principally by transferring salinity to the North Atlantic region from the North Pacific as described in. Edwards and Marsh (2005) (although we deviate from this flux pattern as described subsequently).

Our implementation of the cGENIE model includes a relatively complete description of the cycling of carbon and oxygen in the ocean plus exchange with the atmosphere, as described in Ridgwell et al. (2007) and Cao et al. (2009). In addition, the carbon isotopic ($\delta^{13}C$) composition of all the carbon pools plus associated fractionations between them are explicitly account for, as described in Ridgwell et al. (2007) and with additional description and evaluation in Kirtland Turner and Ridgwell (2016). Given the paucity of constraints on the evolving patterns of aeolian fluxes to the ocean surface, we omit a marine iron cycle, and include only a single nutrient, phosphate, as potentially limiting to biological productivity in the ocean in our configuration. We also employ an idealized organic matter export scheme and hence biological carbon pump (Cao et al. (2009)) and do not attempt to simulate ecosystem composition and dynamics (as per for example in Wilson et al. (2018)).

In the original modern configurations of the cGENIE model, ocean bathymetry and land-sea mask grids, together with files supporting the calculation of ocean circulation files (defining 'islands' and circulation paths around islands), are derived from filtered global topographic observations (ETOPO5) (Edwards and Marsh, 2005). Furthermore, because the atmospheric EMBM component lacks clouds and dynamics (e.g. winds), additional spatial fields must be provided as fixed (annual average) boundary conditions. Firstly, a zonally-averaged planetary albedo profile is applied (originally as a simple

cosine function of latitude as per Edwards and Marsh (2005)). Secondly, fields for: (a) vectors of wind stress on the ocean surface, (b) vectors of surface wind velocity in the atmosphere, and (c) short-term wind speed, are derived from modern observations and applied respectively to: (a) driving surface ocean circulation, (b) transporting heat and moisture in the atmosphere and for calculating heat and moisture exchange between ocean surface and atmosphere, and (c) in calculating air-sea gas exchange.

For this study, we create a series of new continental and surface boundary condition configurations of the cGENIE model for the 7 time-slices spanning the late Holocene through to mid Miocene. As per previous (deeper time) paleo applications of the cGENIE.muffin model (e.g. Ridgwell and Schmidt, 2010), rather than observations, we derive the required boundary conditions from a representative fully coupled GCM experiment. To create the Miocene slices boundary conditions, we re-grid the GCM HadCM3LM2.1aE (hence referred to as HadCM3L) model configuration to the cGENIE grid using the

method described in Appendix B, including bathymetry, wind stress and velocity, and planetary albedo. Due to the re-gridding process, this results in an open Central American Seaway from 15 Ma up to and including present (late Holocene). To test for the role of the closing of the CAS, we create a parallel series of an additional 5 time-slices spanning 10 Ma to present, in which we force a fully closed CAS in the cGENIE model grid. (Although the CAS is not open today, we include an open CAS for the late Holocene time-slice for completeness and to test for the ability of benthic $\delta^{13}$C observations to distinguish between the

two alternatives.) The resultant cGENIE bathymetry for each time-slice is shown in Fig. 3 for all 12 (7 CAS open and CAS 5 closed) continental configurations, with a zoomed in section around the CAS and a cross section of the shallowest CAS depths for each open CAS configuration. We compare our PLATES derived paleolocations to those derived from the HadCM3L in supplementary Fig. B1.

    The impact on ocean circulation of changing grid resolution and swapping the HadCM3L physics for that of the

cGENIE model is included in supplementary Fig. B2 in the form of a comparison of modern (late Holocene) simulations. HadCM3L generally shows much stronger currents such as the Antarctic Circumpolar Current (ACC) than does cGENIE, with the finer resolution HadCM3L enabling features like the western boundary current in the Atlantic to be reasonably resolved whereas it appears weak and diffuse at much lower resolution in cGENIE. In addition to the lower resolution model grid employed in this study, the frictional geostrophic approximation physics employed in cGENIE (Edwards et al. 1998) also tends

to dampen the wind-driven circulation. However, the fidelity of the large-scale patterns of ocean circulation in cGENIE are sufficient to support the simulation of anthropogenic carbon uptake and deep ocean radiocarbon on par with other 3D ocean circulation based Earth system models (Cao et al. 2009) as well as generally plausible distributions of dissolved nutrients and oxygen (e.g. Crichton et al. 2021).

## 2.3 Model experimental design

In a 'perfect' paleo model (which does not exist) and under the correct greenhouse gas boundary conditions (which are poorly constrained), climate and ocean circulation would be a correct and emergent property of the model and provide an exact (within proxy measurements and calibration error) match to the data. Rather, the premise of our experiment design and study is that we can find a specific combination of climate state and pattern of global ocean circulation that reasonably accounts for the observed distribution of ocean temperature proxies and benthic $\delta^{13}C$. In cGENIE, we consider $CO_2$ as a primary uncertainty

in the model – not only in terms of uncertainty in the real past value, but additionally in the radiative forcing required in cGENIE to generate appropriate ocean temperatures. Additionally, because there is no prior expectation that for a 'correct' surface climate the emergent state of ocean circulation is at-all correct in cGENIE, we consider the salinity flux adjustment (hereafter 'FwF') as an additional 'unknown' and moreover, as a means-to-an-end in adjusting basin (to global) scale patterns and strength of circulation (and specifically Atlantic overturning circulation). By varying both controlling factors (parameters),

we thereby seek to find the circulation pattern and climate (temperature) state that best reproduces the data. It should be noted that for this study, we ignore the strength of the biological pump as an additional and independent control on benthic $\delta^{13}C$ and in our particular model configuration, remineralisation rates of organic matter in the ocean interior are not dependent on local temperature. Similarly, we assume the modern ocean nutrient (phosphate) inventory throughout all time-slices.

     For each of the 12 (7 CAS open and 5 CAS closed) time-slices and model configurations, we carry out a 2D parameter

space ($CO_2$ vs. FwF) sweep via an ensemble of model experiments. In each ensemble, we test (radiative forcing equivalent) atmospheric $CO_2$ values of 280, 400, 560, 800, 1120 and 1600 ppm, and vary the salinity flux correction applied to the North Atlantic (FwF) between 0.0 Sv and 0.7 Sv, in increments of 0.1 Sv, for a total of 48 members in each ensemble. The $CO_2$ values are chosen either as simple multiples of 280 ppm (pre-industrial / late Holocene), or commonly assumed GCM values (e.g. the 400 ppm in Farnsworth et al. (2019)) and multiples there-of. We chose the range of FwF values to encompass the

equivalent tuned modern cGENIE model value of 0.32 Sv (Edwards and Marsh, 2005). Furthermore, because the land-sea mask progressively changes across the 7 time-slices, we create a common mask for the purpose of salt transfer. We identify all the points in the grid that are ocean ('wet' points) across all configurations and lie northwards of the northernmost extent of the CAS and add salinity to these, we identify all the grid points outside of the North Atlantic that are ocean in all configurations, and remove salinity from these. The total FwF value is then divided evenly across all (equal area) grid points

in the North Atlantic (positive salinity input) and 'elsewhere' (negative salinity input) such that global ocean salinity is always conserved. The mask is shown in supplementary Fig. B3.

     For each ensemble member and each time-slice (a total of 48×12 = 576 model simulations), we spin-up the cGENIE model for 10,000 years. In the absence of appreciable inter-annual variability and unlike in fully coupled GCM experiments, multi-decadal averaging is not necessary in the cGENIE model and we take the last annual average (year 10,000) of the

simulation in order to carry out the model-data comparison.

The one caveat and complication to how we conduct the model-data comparison is that in running the parameter ensembles, we identified self-sustained oscillations in global ocean circulation in a subset of the simulations that affected both mean benthic temperature (up to several °C) and $\delta^{13}$C (several tenths of a ‰). These oscillations were of varying period and magnitude and occurred only in simulations with sea-ice cover present (and hence in the lower range of eq.$CO_2$ forcing), and generally for the low-to-mid range of FwF values (see Appendix C and supplementary Fig. C1 for more details). Whilst this raises extremely interesting questions about past ocean circulation dynamics, it is not the focus of this study (but will be followed up in a subsequent study). Here we need to identify a 'representative' ocean state of the model in order to carry out the model-data comparison consistently. To create a representative state for each $CO_2$-FwF combination, we therefore parsed the ensemble cGENIE model output, identifying ensemble members characterised by self-sustained oscillations. For the these ensemble members, we identified the period of the oscillation and averaged the model output over one full period, starting the average from the end of the 10,000 year spin up and working back (towards the start of the model experiment). Both unmodified run-end and reconstructed mean annual averages were then treated exactly the same in terms of carrying out model-data comparison.

## 2.4 Model-data comparison

The collated datasets were used as an observational constraint to determine which combination of $CO_2$ and North Atlantic salinity flux correction (FwF) produces the best-fit ocean state to the data. This is performed quantitatively by statistically comparing local model output with the data points for all three datasets, and combining these to produce a final "best-fit" model setting for each time-slice. The statistical methods applied are: 1) the difference between the mean of the dataset and the mean of the model output at the data-locations, producing an offset value or mean bias, 2) an overall measure of goodness of fit of the model to the dataset known as "M-score" (Watterson, 1996). Where several data-points are located within one model grid square, the mean of the data values is used to compare with the model value. For the benthic data, each data-point is assumed to be on the ocean floor, so the model's bathymetry determines the data depth, and the data value is compared with the model value in the deepest water in that location.

## 3 Results

### 3.1 Global mean and spatial data constraints

We start by assessing the model ensemble members against global mean observational constraints. In Figure 4, we show the global mean benthic temperature data estimate from Cramer et al. (2011) plotted on top of the modelled global mean benthic (sea-floor arithmetic mean) ocean temperature for our ensemble in Fig. 4. Both $CO_2$ and FwF have a strong effect on benthic temperature. Bathymetry, albedo and windfields (which differ in our time slices, see Appendix B) also have a direct effect on benthic ocean temperature, with 15 Ma having a tendency to a warmer deep ocean compared other time slices even for the

same $CO_2$ and FwF in the open CAS cases. A range of combinations of $CO_2$ and FwF would satisfy the global mean benthic temperature constraint (shown as the dashed white line, with range as thin white lines). The FwF affects North Atlantic surface salinity and therefore has an impact on thermohaline circulation; higher FwF values create more saline surface North Atlantic

waters of lower buoyancy, transporting more of the surface heat to the deep ocean. This results in combinations of higher $CO_2$-low FwF, or lower $CO_2$-high FwF being equally possible to satisfy the benthic global mean temperature constraint, but with a general trend from 15 Ma towards the Holocene of reducing $CO_2$ and FwF (as the dashed line moves towards the top left in Fig. 4) for both the CAS open and CAS closed cases. For almost all slices, an elbow-type shape is present in the modelled benthic temperatures. As $CO_2$ reduces for a given FwF, global mean benthic temperature is either stable or even increases at a

certain point (Fig 4). At $CO_2$ levels lower than this "elbow", the cooler surface ocean favours the sinking of waters in high latitudes, supporting the thermohaline circulation. At 12.5 Ma the elbow turning-point $CO_2$ value is around 800 ppm and this gradually reduces to about 400 ppm in the Holocene (for mid-range FwF). At $CO_2$ higher than this elbow, there is only a very weak (or shallow) AMOC. In the closed CAS cases, this elbow is less evident than for the open CAS, which may suggest that an open CAS results in a climate more (or differently) sensitive to $CO_2$ changes than a closed CAS, via AMOC.

The results of the M-score spatial statistical comparison (where the closer to 1, the better the model can simulate the data) for the three individual data-sets and all model ensemble members are shown in Figure 5, where for each eq.$CO_2$ forcing, the FwF from 0.0 to 0.7 is plotted sequentially.

                The surface temperature constraint is rather insensitive to FwF, with eq.$CO_2$ being the main controller on the fit to data. Whether the CAS is open or closed also has very little effect on the ability of the model to reproduce the data. The highest

M-scores indicate that a general fall in eq.$CO_2$ forcing results in best fit to the surface temperature data, with generally higher M-scores achieved in the more recent time-slices. The 12.5 Ma and 15 Ma have low scores (less than 0.4) compared to other time-slices for surface ocean temperature, indicating that even at 1600ppm the model cannot reproduce the data-indicated surface ocean temperatures very well.

                For the benthic temperature dataset, as with the global mean ocean temperature (Fig. 4) both eq.$CO_2$ and FwF control

deep ocean heat distribution, with highest M-scores again generally showing a reducing $CO_2$ and reducing FwF towards the present. The CAS-open case results in higher M-scores (i.e. better fit to data) than CAS closed, even for the Holocene. As we know the CAS is closed in the Holocene, ~~so~~ these higher M-scores may therefore be the result of a model-bias.

                The $\delta^{13}C$ M-scores for the Holocene show higher values for a CAS closed than CAS open, indicating $\delta^{13}C$ may be a better indicator than benthic temperature for whether CAS is open or closed in our cGENIE model configurations. For the 10

Ma slice, for a higher $CO_2$ value, a closed CAS better fits benthic $\delta^{13}C$ data, and if a gradually reducing $CO_2$ toward the present is assumed (from the surface temperature data), a closed CAS generally results in higher M-scores for all other timeslices as well. A shift occurring in the patterns of benthic $\delta^{13}C$ can also be seen in the raw data. The latitudinal trend for $\delta^{13}C$ data (dotted lines) in Fig. 2 shows that the Atlantic appears to switch trend between 12.5 Ma and 10 Ma. Prior to 12.5 Ma the North Atlantic data indicates a generally more negative $\delta^{13}C$ than in the South Atlantic – a situation that reverses from 10 Ma onwards

when more positive $\delta^{13}C$ tend to occur in the North Atlantic (although $R^2$ is low). In the Pacific Ocean, the $\delta^{13}C$ gradient from

south to north tends to intensify towards the present, with the largest gradient seen at 2.5 Ma. Overall, the range of $\delta^{13}$C values increases from 15 Ma towards the present. (Note that the colourbar values change, but the range (difference between the minimum and maximum value) is 1.8‰ for all timeslices in Fig. 2.)

## 3.2 Combining the constraints

All data constraints together are shown in Fig. 6, with the surface and the deep ocean temperature constraints (showing both M-scores and where bias in deep ocean temperature is less than 2°C), plus the global mean benthic temperature estimate are mapped onto the statistical fit surface (M-score) of model vs. observed $\delta^{13}$C. We mark the best-fit selection with a star and include a range of estimates delineated by error bars on the best fit $CO_2$ and FwF values. Note that the best-fit is only identified from those specific values for which we ran the model ensemble members (i.e. we do not interpolate between settings).

It is clear from this that the primary constraint on atmospheric eq.$CO_2$ in the model in this study is the proxy reconstructed surface ocean temperature. This is much less sensitive to the FwF value than the other benthic ocean constraints. This is because surface ocean circulation patterns are largely dictated by the wind stress forcing, that we do not vary within any single time-slice. Given a surface ocean circulation pattern that is relatively immune to changes in the applied FwF, the ensemble member eq.$CO_2$ value, and with it the surface climate state, control the mean and pole-to-equator gradient of SST
and hence model-data surface temperature fit. For all the following analyses of the model-data fit, we hence start with consideration of SSTs and the choice of eq.$CO_2$ for each time-slice in turn, then bring in additional constraints and consideration of the value of FwF.

        A priority for selecting good combinations of eq.$CO_2$ and FwF is to ensure that the global ocean heat distribution shows reasonable agreement with our proxy temperature data. The benthic global mean temperature estimate of Cramer et al
(2011) is then used as an ancillary guide to our benthic temperature set derived from $\delta^{18}$O and the salinity correction, and in most cases these show quite good agreement with each-other (with 15Ma and both the Holocene cases showing lowest agreement; N.B. the Cramer et al (2011) estimate applied for the Holocene actually represents a mid-climate state rather than the interglacial, which explains the difference there). Its should be noted that the global ice volume-linked global $\delta^{18}O_{sw}$ that we use in our temperature calculation is derived from the Mg/Ca temperature dataset from Cramer et al (2011) (Table S2), so
these are not fully independent datasets. The M-score for the $\delta^{13}$C is used to determine the direction of any adjustment needed when SST and benthic temperatureT constraints have low agreement for the best eq.$CO_2$-FwF combination, especially for particularly high $\delta^{13}$C M-scores (for example at 10Ma open-CAS, where benthic temperature suggest a FwF of 0.1 Sv, and eq.$CO_2$ of 1120ppm, but the $\delta^{13}$C M-score strongly increases in the direction of a lower $CO_2$ and higher FwF). More details on the selected combinations are available in Appendix D. All the selected settings for eq.$CO_2$ and FwF and their ranges are
summarised in Fig. 7 (and marked on Fig. 5), together with some recent data estimates of atmospheric $CO_2$. Also plotted in Fig. 7 are global mean ocean surface and benthic temperatures, and sea ice extent and thickness for each time slice, showing the modelled evolution since 15Ma to the Holocene.

Modelled surface ocean temperature drops with the falling eq.$CO_2$; deep ocean (benthic) temperatures also fall, but show some dependence on FwF at the same time (Fig 7). In the highest FwF case – 4.5 Ma CAS open – despite a halving of eq.$CO_2$ compared to 7.5 Ma, deep ocean temperatures are only slightly lower than 7.5 Ma. This is due to the increased salinity N. Atlantic waters, and the cooler polar temperatures (demonstrated by the presence of more sea-ice) promoting sinking of these surface waters to the deep, delivering relatively more surface heat to the deep ocean at 4.5 Ma than at 7.5 Ma. Conversely, in the 2.5 Ma CAS-open case, FwF is lower than for 4.5 Ma but eq.$CO_2$ is the same. Here, less of the surface heat is delivered to the deep ocean, resulting in a cooler deep ocean, but a warmer surface ocean and lower sea ice than at 4.5 Ma (for the best fit setting). Sea ice is present from eq.$CO_2$ levels of 800ppm from 10 Ma, generally increasing both in extent and thickness with falling eq.$CO_2$ (with the exception of the previous example), whether the CAS is open or closed.

As a note, the imposed eq.$CO_2$ does not change due to changes in ocean carbon distribution or carbon exchange with the atmosphere; we apply a flux-correction such that atmospheric $CO_2$ is always restored to the eq.$CO_2$ we initially impose, therefore we do not attempt to attribute the source of $CO_2$ decline since 15Ma to any particular mechanism.

### 3.3 Atmospheric and global mean benthic carbon-13 ratios

Using the $CO_2$ FwF settings identified from the combined constraints, we can derive model-predicted estimates of the trends in atmospheric $\delta^{13}CO_2$ and of global mean benthic $\delta^{13}C$ over the past 15 Ma. In the ensembles, the model atmosphere is forced with a $\delta^{13}CO_2$ value of -6.5‰ for all simulations, so by determining the mean bias between the modelled benthic $\delta^{13}C$ points, and the data benthic $\delta^{13}C$ points we can derive the $\delta^{13}CO_2$ value that would produce the lowest overall model-data error (so, essentially we inverse-model the $\delta^{13}CO_2$ value). This is shown in Fig. 8. This diagnosed history of atmospheric $\delta^{13}CO_2$ can be used as a means of identifying changes in the global carbon cycle (e.g. Hilting et al., 2008) or as initial condition values for future model (and model-data) based studies.

Similarly, using this model-data benthic $\delta^{13}C$ bias we can derive the global surface and benthic mean $\delta^{13}C$ trend using the local bias of our model to the benthic $\delta^{13}C$ data. This global mean deep ocean $\delta^{13}C$ is shown in Fig. 8 plotted along with the benthic $\delta^{13}C$ stack from Westerhold et al. 2020. The datapoints from our benthic $\delta^{13}C$ dataset are also shown on Fig. 8, which tend to be higher (especially in younger time slices) than the benthic stack data; our benthic dataset is not limited to only low and mid latitudes (see Fig. 2) unlike the benthic stack data. The N.Atlantic datapoints have generally more positive $\delta^{13}C$ than other regions and a large proportion of our datapoints are from this region. W~~However, w~~e use local model-data benthic $\delta^{13}C$ differences to constrain the global benthic $\delta^{13}C$ mean, but still find this mean to be on the higher (heavier) end of the benthic $\delta^{13}C$ stack from Westerhold et al 2020. The modelled surface and deep ocean mean $\delta^{13}C$ falls by around 0.8 ‰ between 15 Ma to the Holocene values, whereas atmospheric $\delta^{13}CO_2$ falls by up to 1.5 ‰ at the same time.

### 3.4 Best fit model-data maps

In general, in the older time slices, cGENIE underestimates temperatures in the North Atlantic (Fig. 9) and this results in a higher M-score for the highest $CO_2$ settings for 15, 12.5 and 10 Ma slices (see fig 5). In these older time slices, the model high

latitude temperatures tend to be too low compared to data – this is an established characteristic of warm climates, where climate models tend to struggle to reproduce the flatter latitudinal temperature gradients seen in data (Goldner et al., 2014). Whether the CAS is open or closed makes little difference to the model-data fit, as our datapoints are not in regions very strongly affected by any resulting change in surface heat distribution. However, the temperature difference map shows that a closed CAS results in a colder modelled S. Atlantic surface ocean than an open CAS, and slightly warmer Pacific surface ocean (note

although the FwF are not the same between the cases, it is the CAS configuration that dominates the surface ocean temperature differences). These temperature differences of 1 or 2 °C in high latitudes would have implications for sea ice and ice shelf growth in the cooler timeslices.

        Taking account of the deep ocean salinity in the temperature calculation from $\delta^{18}$O data makes a significant difference to deep ocean temperature patterns (Fig. 10). When flux corrections are higher (mainly from 10 Ma), the saltier North Atlantic

waters are less buoyant, and more readily sink. This higher salinity results in higher calculated temperatures (as $\delta^{18}$O$_{sw}$ is corrected for local salinity), with data-points in shallower waters most strongly affected (see Appendix A). Accounting for water salinity increases some data-points temperature by more than 3°C (Fig. 10) in the N. Altantic. The inverse is true for data locations in the deep Pacific, where some temperatures are corrected lower. In both cases this affects the model-data fit, likely resulting in higher $CO_2$ and/or higher FwF combinations than would otherwise be the case it we had conducted the

model-data comparison using only uncorrected $\delta^{18}$O derived temperatures. The CAS-open case for the Holocene shows higher M-score than CAS closed for the model fit to benthic temperature~~T~~ (which we know should not be the case as the CAS is actually closed in the Holocene) (Fig 5). The Indian ocean appears to be a location where modelled benthic temperature~~T~~ is too high compared to our $\delta^{18}$O derived data, whilst the Pacific is generally too low (and where the CAS-open case shows a better fit to data). The 2.5 Ma modelled benthic temperature is slightly too high, and this is probably due to the 400ppm forcing,

where 280ppm would also have been a good fit (see Appendix D). The two oldest timeslices show modelled benthic temperature~~T~~ with a much smaller range than data indicates, which may be a result of assuming all benthic datapoints are located on the ocean floor in the cGENIE model grid (where ocean bathymetry is smoothed out compared to the real-world case). In general, a closed CAS results in a relatively warmer deep Atlantic and cooler Pacific than an open CAS for the best fit settings.

The range of values of modelled $\delta^{13}$C are in general agreement with the data-indicated range (Fig. 11). At 15 Ma the modelled higher $\delta^{13}$C values in the high southern latitudes are not clearly shown in the data, particularly affecting the Indian Ocean. The 12.5 Ma slice shows very low correlation for the distribution of $\delta^{13}$C. T~~, t~~his may reflect the large climate and/or circulation changes occurring at the time (the ~~m~~Mid-Miocene Climate Transition, MacKensen and Schmeidl 2018) and the fidelity of the complied dataset especially with respect to age, or again perhaps partly due to the smoothed model bathymetry.

From 10 Ma to the Holocene all the CAS closed cases show better fit to data than CAS open (see also Fig. 5) (the linear regression for each is shown in the crossplots, the closer this line is to diagonal or the 1:1 line, the better the fit to data). Importantly, the model-data benthic $\delta^{13}$C correlation for 0 Ma is notably better for CAS closed than open. This can also be

seen in Figure 5 where regardless of the parameter combination, a better M-score for CAS closed is always obtained (with the exception of a single ensemble member for 0 Sv FwF and 280 ppm $CO_2$).

Every ensemble member, regardless of the specific atmospheric $CO_2$ assumption, was driven with a $\delta^{13}CO_2$ value of -6.5‰. The model output colour-shading in Fig. 11 is according to this value so that each model time-slice can be compared with any other; the colour-shading indicates deep ocean $\delta^{13}C$ relative to a universal atmospheric $\delta^{13}CO_2$ value (the data indicated actual benthic $\delta^{13}C$-indicated scale is shown to the left of each colourbar, and the scale for $\delta^{13}CO_2$ at -6.5‰ is to the right of each colourbar). As a general rule, at the warmest earlier periods the surface ocean southern high latitudes signal

dominates the deep ocean from the south, and as time progresses and the climate cools the surface N. Atlantic becomes dominant from the deep Atlantic. This is accentuated for the CAS closed cases.

### 3.5 Best-fit ocean cross-sections

The modelled Pacific and Atlantic ocean $\delta^{13}C$ cross sections are presented in Fig. 12 (note that for benthic $\delta^{13}C$ datapoints, we

always assume they are at the modelled sea-floor, and so they are not shown here). These are created following the transect shown on the inset map in Fig. 12 and averaged over a width of 3 model grid squares, so a 30° swath, the same longitudes for each timeslice. Again the colour shading is with respect to a $\delta^{13}CO_2$ of -6.5‰ to be able to compare timeslices as for Fig. 11. In the older slices, there is a larger offset between whole ocean $\delta^{13}C$ and atmosphere than in the younger slices (the ocean is relatively heavier in $\delta^{13}C$ in older time slices). The Pacific Ocean becomes progressively more negative (lighter) in $\delta^{13}C$

through time, accentuated by a closed CAS. The mid-depth mid-latitudes Pacific appears sensitive to the closing or opening CAS, with modelled $\delta^{13}C$ indicating some 0.2 to 0.5 ‰ difference between the open or closed CAS.

        The modelled barotropic and global streamfunctions for the best-fit settings are shown in Fig. 13. Surface wind fields derived from HadCM3L exert some control on the barotropic streamfunction, as does thermohaline circulation and bathymetry. When the CAS is open a circum-S. America circulation is in operation, allowing the mixing of Pacific and Atlantic water

masses. The Antarctic Circumpolar Current (ACC) is already well established by 15 Ma, and strengthens towards 2.5 Ma with cooling and with the northern movement of S. America (so widening of the Drake Passage). At the Holocene the ACC is weaker compared to 2.5 Ma, this may be a similar effect to the weakening of the ACC in cold glacials relative to interglacials (Toyos et al 2020), and may possibly be related to increased sea ice extent at the lower eq.$CO_2$ Holocene 280ppm compared to the 2.5 Ma 400ppm forcing, among other things (such as bathymetry and windfield).

The global streamfunction shows the increasing dominance of northern hemisphere deep water through the cooling from the Miocene to the Holocene, with a gradually deepening and strengthening AMOC. This is the case for both CAS open and CAS closed sets, but with CAS open requiring a more salty N. Atlantic to induce it than for CAS closed. The strongest AMOC is achieved in the coldest Holocene timeslice, with eq.$CO_2$ of 280ppm, cool poles and largest sea ice extent and thickness.

## 4 Discussion

Overall, we find that global surface and deep-ocean cooling since the mid-Miocene can be accounted for in cGENIE by the combined effects of changing bathymetry, eq.$CO_2$ forcing, and salinity of the N. Atlantic ocean. The middle Miocene eq.$CO_2$ required in the model is relatively high (1120 ppm), and a small (0.1 Sv) salinity forcing of the N. Atlantic can fairly well reproduce surface to deep ocean heat distribution. However, latitudinal temperature gradients are not well reproduced, with high latitudes too cool compared to data in the warmer time slices. The eq.$CO_2$ forcing reduces steadily from the mMid-Miocene to the present to satisfy surface temperature data, while the required salinity forcing of the N. Atlantic increases to a maximum in the late Miocene/early Pliocene, then reduces again towards the present. Our modelled closed CAS from 10 Ma to present all show better agreement with $\delta^{13}C$ data distributions than the open CAS cases (due to data $\delta^{13}C$ Atlantic Ocean latitudinal gradients trends in Fig 2. we did not simulate a closed CAS at 12.5 or 15 Ma).

In the next sections, we compare findings from this study against published evidence for the plausibility of the climate forcings and conditions we have identified.

### 4.1 Heat distribution and eq.$CO_2$

Previous climate modelling studies of the Miocene have generally found that proxy data for an atmospheric $CO_2$ concentration around 400pm is insufficient to explain the observed warmth (Greenop et al 2014; Goldner et al., 2014; Bradshaw et al 2012; Krapp and Jungclaus, 2011). Estimates of Miocene $CO_2$ in more recent studies have trended higher compared to the earlier estimates (e.g., ~220ppm at 15 Ma, Pagani et al., 1999). Our required eq.$CO_2$ forcing in the oldest, 15 Ma timeslice is 1120 ppm. This compares with: 450-550ppm using a multimethod multitaxon $pCO_2$ reconstruction (Stenthorsdottir et al 2020b), 470 to 630 ppm from a boron $CO_2$ proxy (Sosdian et al., 2018); 528 ppm or 912 ppm from Neotropical fossil leaf assemblages (Londoño et al 2018); or in the region of 1000 ppm in a recalculation of the alkenone $\epsilon_p$ $CO_2$ proxy (Stoll et al 2019). However, dDespite our eq.$CO_2$ forcing being higher than these recent estimates, we still obtain a low M-score for the surface ocean temperature distribution.

A portion of the $CO_2$ forcing model-data discrepancy may be explained as the contribution of other greenhouse gases to climate forcing during the Miocene. In the configuration of the cGENIE model used here, only $CO_2$ is imposed as a greenhouse gas, so the imposed atmospheric eq.$CO_2$ concentration should be understood as an 'equivalent' climate forcing encompassing other greenhouse gases. Wetlands are a significant source of atmospheric methane in the present day, as well as representing large terrestrial carbon stores. Extensive wetlands existed during the middle Miocene (Eronen and Rossner, 2007; Hoorn et al., 2010; Morley and Morley, 2013), with the generally warmer wetter climate and lower elevation topography being favourable to their initiation and persistence. These conditions would be conducive to elevated methane production in the past (Dean et al., 2018). A large Antarctic ice sheet has been identified for the period soon after the Miocene climatic optimum (Hautvogel et al., 2012; Badger et al., 2013; Pierce et al., 2017), which would have caused a fall in sea-level. Gas hydrates were found to have likely destabilised during the sea-level lowering of the Miocene in the (present-day) Appenines in Italy

(Argentino et al., 2019) and in the Black Sea Basin (Kitchka et al., 2016), that may have had a transient impact on atmospheric methane levels and climate fluctuations. A larger contribution of methane to eq.$CO_2$ forcing in the Miocene relative to the Holocene is certainly plausible (atmospheric chemistry and baseline methane levels also affect the lifetime of atmospheric methane, Schmidt and Shindell, 2003).

During the warmest intervals – 12.5 Ma and 15 Ma – we find our largest differences in model to data surface temperature in the N. Atlantic (Fig. 8), with modelled N. Atlantic temperature being much lower than sea surface temperature (SST) indicators. The model overestimates the latitudinal temperature gradient in the warmer climates, with heat transport to the poles likely being too low. Increased ocean heat transport was found to reduce the latitudinal temperature gradient and the location of Hadley and Ferrel cells in a modelling study by Knietzsch et al (2015). Changes in land surface cover and increased northern transient eddy atmospheric heat transport (linked to storm tracks in mid-latitudes) were found to increase heat transport to high latitudes in a modelled late Miocene in a fully coupled AOGCM by Micheels et al. (2010). These indicateing the importance of atmospheric circulation, which in the 2D energy-moisture balance atmosphere configuration used here, is simplified and invarianthighly idealized.

Equilibrium climate sensitivity is almost the same in all our modelled climates, with a value of 2.9 (±0.05 at 1σ and a range of 0.174). However, in more complex models, the interaction between vegetation and paleogeography has been shown to give rise to a higher climate sensitivity in Miocene climate modelling (Bradshaw et al., 2015; Micheels et al., 2010), with warming occurring independent of $CO_2$ increase due to vegetation and lower-elevation topography (Henrot et al. 2010) and their atmosphere feedbacks (Knorr et al. 2011). Indeed, chemistry-climate feedbacks linked to vegetation were found to be as strong as or more important than $CO_2$ forcing also in the Pliocene (Unger and Yue, 2013). The significant changes occurring in global vegetation distribution since the middle Miocene (Pound et al., 2012) then may be critical to fully reproducing observed SST patterns. The absence of any representation of vegetation and associated feedbacks in the version of cGENIE we employ may then help account for the low M-scores in model-data fit at high mid-Miocene $CO_2$ forcing of 1120ppm, and in as well as some of the discrepancies between model eq.$CO_2$ and data indicated $CO_2$ in the older timeslices (Fig. 7). A higher climate sensitivity in the Miocene, and vegetation-atmospheric feedbacks, and increaseding heat transport to the poles may therefore be able to improve the model-data fit.

Miocene Climatic Optimum (MCO) proxy data suggest global surface temperatures 7 to 8 °C warmer than modern (Steindthorsdottir et al. 2020). Our modelled mid Miocene is around 6 °C warmer than the Holocene but in response to a relatively high estimate of eq.$CO_2$ forcing – 1120 ppm. The Mg/Ca thermometer estimates deep ocean temperatures around 6 to 7°C warmer than present (Cramer et al. 2011, Lear et al. 2015), where we model temperatures around 3 to 4°C warmer (as a deep ocean global mean). Indian ocean benthic temperaturesT were found to be 9°C warmer than present at the Miocene Climate Optimum, (and 6°C warmer than present after the Miocene Climate Transition at 14.7 to 13 Ma) by Modestou et al. (2020), far warmer than we model (Fig. 10). If high latitude surface waters were warmer in the model (as data suggests), this would not only raise the global mean (and possibly allow for a lower eq.$CO_2$ forcing), but would drive a globally warmer deep ocean, as meridional overturning circulation transports high latitude warm waters to the deep and which may then explain

some of the disagreement in surface and deep ocean T constraints at 15Ma (Fig. 6). However, it would probably not raise the deep Indian ocean temperatures enough in line with the recent estimates from Modestou et al. (2020), suggesting a possible model circulation bias for the Indian Ocean at the mid Miocene.

The feedbacks affecting climate sensitivity and the latitudinal temperature gradient are likely climate-sensitive, affected by topography, and perhaps also directly ~~by~~ temperature-sensitive. ~~These parameters~~Topography, climate and temperature approach those seen in the present ~~since 15 Ma~~over each successive time slice, suggesting that the~~se~~ climate feedbacks that we do not capture ~~model~~may become less important with each time-slice as we approach the Holocene.

## 4.2 North Atlantic salinity and mountain uplift

The salinity flux correction that we apply to the N. Atlantic can be considered as primarily representing an incompletely 495 resolved runoff pattern across N. America, itself arising from (poorly resolved) atmospheric moisture transport and (unresolved) topography. In a warm, higher $CO_2$ climate like the Miocene, the atmosphere is able to hold more water vapour and the result is a globally wetter climate. However, the North American plains saw the rise of grasslands that favour a drier climate (Janis et al., 2002) through the Miocene, and this is linked to the uplift of Mountain ranges in the west that created a rain shadow on the central plains. Over the last 12 million years in the Sierra Nevada, a rain shadow similar to present was 500 identified by Mulch et al. (2008), indicating mountain uplift in this area occurred prior to 12 Ma. Further north in the Cascade Mountains, high uplift and erosion rates were dated to late Miocene (12 to 6 Ma), and the Coast Mountains and British Colombia uplift from 10 Ma (Reiners et al. 2002). ~~We thus expect that~~ T~~t~~he flux correction implicitly includes the impact of mountain building from around 12.5 Ma to 6 Ma in addition to changing atmospheric moisture content (and after 6Ma, a global cooling reduces the energy in the hydrological cycle). Whether the modelled CAS is open or closed has an effect on the 505 required FwF, with a closed CAS reducing the required salinity correction by isolating the Atlantic from the Pacific water masses and affecting N. Atlantic water buoyancy. Our best-fit FwF increases from 15 Ma, with highest values seen at 4.5 Ma for CAS open and 7.5 Ma for CAS closed (Fig. 7), so generally in agreement with evidence of the timing of N. American mountain building and changing atmospheric moisture capacity, and with CAS-open requiring a higher FwF than the CAS-closed cases.

## 510 4.3 Ocean gateways and circulation

The Antarctic Circumpolar Current (ACC) strengthens through time up to 2.5 Ma in our best fit model timeslices, with an ACC clearly already established at 15 Ma, when the Drake Passage was already open and fairly deep (Fig. 3, Fig. 13). Some evidence suggests that a volcanic arc may have inhibited ACC flow until after the mid-Miocene (Dalziel et al. 2013) or even later (Pérez et al. 2021), although other studies find an earlier onset for the ACC (e.g. Pfuhl et al 2005). However, even if a 515 volcanic arc was present, the upscaling to the cGENIE grid would smooth out this feature. A full consideration of changing bathymetry and gateways and how this impacts the ACC, Antarctic Bottom Water production and properties, and any knock-on effect on global climate, would need to be the subject of a separate study.

Evidence shows a global cooling and an increase in ice growth at the MMCT (mMid-Miocene Climate Transition occurring 14.5 Ma to 12.5 Ma), which we do not identify in our 12.5 Ma timeslice. The MMCT has been linked to the closing

of the eastern gateway of the Tethys Sea, which in our model set up is already closed at 15 Ma. An open eastern Tethys gateway allowed Tethyian Indian Saline Water (TISW) flow into the Indian Ocean that may have inhibited east Antarctic sea ice growth, until its closure at 14 Ma (MacKensen and Schmiedl, 2018). Surface North Atlantic temperature data lends some support to an enhanced AMOC or strengthened North Atlantic Current during the MMCT (and that heat distribution patterns are not consistent with solely $CO_2$-driven cooling in this location) (Super et al 2020). Furthermore, for the 12.5 Ma timeslice, our $\delta^{13}C$

dataset provides only a very weak constraint on the forcings. It is certainly possible that with a better (less noisy) $\delta^{13}C$ dataset, the 12.5 Ma best-fit could target a lower-$CO_2$ (so more in line with $CO_2$ proxy data as Fig. 7) and higher FwF, with a more saline N.Atlantic very possibly linked to a salty-water contribution from the Tethys sea that is now closed off from the Indian Ocean. However, this lower $CO_2$ would then result in lower modelled surface ocean temperatures in the low latitudes, which at 1120ppm are already slightly too low in cGENIE according to data (Fig. 9).

We identify a possible early CAS closure/restriction from 10Ma according to the combined temperature and $\delta^{13}C$ constraint. A wash-house (warm and wet) climate was identified in Europe in certain periods of the Miocene, from 10.2 Ma to 9.8 Ma and from 9.0 Ma to 8.5 Ma, which appeared to be linked to temperatures in the deep N. Atlantic Ocean temperatures (Böhme et al., 2008). This was in-turn linked to a possible temporary restriction of the CAS and greater northward heat transport. A middle Miocene, (even if temporary (Jaramillo et al., 2017)), closure of the CAS was identified by Montes et al.

(2015), further supporting our findings. The CAS is thought to have been definitely closed by 2.5 Ma (O'Dea et al 2017) and evidence shows that some flow between Pacific and Atlantic was still occurring up to that point (O'Dea et al 2017, Jaramillo et al 2017). A CAS closed from 10 Ma shows better agreement to $\delta^{13}C$ data in our study, but we do not consider a restricted CAS, only either open or closed (Fig. 3 shows CAS depths of at least 1000m for our open CAS cases); periodic CAS closing or restriction since 10Ma up to 2.5 Ma is consistent with our findings. A shallow CAS still allowed the formation of deep water

in the Miocene North Atlantic in a GCM model study by Nisancioglu et al. (2003), and even an open CAS in this study (at >1000m, Fig. 3) does not impede the establishment of the AMOC if the surface North Atlantic is sufficiently saline. Our modelled global overturning circulation shows a gradual increase in dominance of northern hemisphere deep water since the mid Miocene, together with the development of the AMOC (Fig. 13). The onset of the AMOC was thought to be a key factor in the expansion of ice at the poles. However, Bell et al. (2015) found that an early Pliocene (4.7 to 4.2 Ma) shoaling of CAS

was had no profound impact on climate evolution, as North Atlantic deep water formation was found to be already vigorous by 4.7 Ma. Vigorous North Atlantic deep-water formation appears to have probably started by 4.5 Ma in our study (Fig. 13), when $CO_2$ drops to 400ppm. A definitive closure of the CAS dates to the late Pliocene (~3Ma, O'Dea et al., 2017). This final Pliocene closure of CAS had been thought to be linked to setting conditions conducive toup conditions for northern hemisphere glaciation, although Lunt et al. (2008) found that an open or closed CAS made little difference to ice sheet size in their model

study of the Pliocene. The closed CAS has a lesser effect on sea ice than does $CO_2$ in our study; sea ice growth intensifies

when $CO_2$ drops to 280ppm in the Holocene timeslice (Fig. 7) for both open or closed CAS, indicating that it is $CO_2$ that would probably drive northern hemisphere glaciation (in cGENIE), rather than a CAS closure.

## 4.4 Benthic Temperatures and the AMOC

The pattern of temperature in the deep ocean arises as a combination of the pattern of ocean surface temperature (and surface
climate in general) and the large-scale circulation of the ocean. In our model, increasing the net salinity transport into the N. Atlantic region induces and strengthens a meridional overturning circulation that in turn drives a warming of the deep ocean, independently of atmospheric $CO_2$ forcing (Fig. 4). Local water salinity has a strong control on $\delta^{18}O$. For example, an increase in measured benthic $\delta^{18}O$ in benthic foraminifera from the N. Atlantic during the late Miocene could be interpreted as evidence of a cooling, or equally it could be attributable to the increased salinity of the sea water, where salinity (rather than temperature)
dominates the $\delta^{18}O$ signal. With the strengthening of Atlantic overturning circulation during the Miocene, the increased salinity of deep N. Atlantic waters exerts a relatively stronger control on temperature calculated from shell $\delta^{18}O$ (than for a weaker AMOC). When accounting for local salinity in the temperature calculation, increases of up to 3°C in some locations are seen when compared to the temperature that is uncorrected for salinity (Fig. 10). Without this correction to the $\delta^{18}O$ temperature calculation, the N. Atlantic would appear to be cooler (and the Pacific warmer), and this would have a large impact on what
the best-fit settings would be for our study (see Appendix A).

## 4.5 Caveats for $\delta^{13}C$ as an ocean circulation tracer

Changes in deep ocean circulation patterns and specifically the aging of water masses and progressive accumulation of isotopically light respired carbon, in theory should be identifiable in global patterns of $\delta^{13}C$ data from benthic foraminifera. However, the processes that control $\delta^{13}C$ are complex, involving both circulation and ocean carbon pumps, which furthermore
are not independent – large-scale changes in circulation that affect nutrient return to the surface will hence also modulate the strength of the biological pump in the ocean. Changing ocean interior temperature patterns may further influence where carbon is respired via a temperature control on the rate of carbon respiration (John et al., 2014), further affecting $\delta^{13}C$. The distribution of $\delta^{13}C$ is also dependent on other factors (such as local water velocity fields and vertical stream functions).

In this study we have applied two of the circulation-linked controllers of $\delta^{13}C$ by applying paleo bathymetry and
altering thermohaline circulation by forcing eq.$CO_2$ and N. Atlantic salinity (FwF). There are uncertainties in the applied paleo bathymetries as well as in the re-gridding method that up-scales to the cGENIE grid (as demonstrated in the re-gridding resulting in an open and fairly deep CAS up to and including the Holocene, an issue equally applicable to GCMs given the relatively small width of the Isthmus of Panama). Regridding also affects the depth at which benthic data is assumed to be located as it smoothes out peaks and troughs in the ocean floor, so some data locations may have been forced deeper or
shallower in the ocean than they would have actually been.

Orbital variations will also have an effect on recorded $\delta^{13}C$ via changes in climate and ocean circulation. Some of the benthic data were high-resolution datasets where it was possible to select a mid-climate value (i.e. between the highest and

lowest values in the orbit oscillation). However much of the data were not of this type, and it may be that the dataset is therefore not well representing the same point in time (in an orbital cycle). This is also generally the case for uncertainties in age models

of the benthic data, and would particularly affect periods in which climate is likely to have been changing over the 2 million year window (that we used to select data) – such as the 12.5 Ma slice. This may explain why $\delta^{13}C$ distributions show very little model-data agreement for any eq.$CO_2$ FwF combination at 12.5 Ma. These effects, combined, create noise in the $\delta^{13}C$ dataset (on top of uncertainties in the data itself), with some time slices showing higher M-scores (in general) than others (Fig. 5).

Despite these limitations, the $\delta^{13}C$ model-data comparison combined with temperature data provided higher M-scores

for more plausible forcings at each time slice, and showed agreement with published work on Miocene to Pleistocene climate and circulation patterns. We identify a likely restricted CAS at 10 Ma and 4.5 Ma, and a closed CAS at 2.5Ma and the Holocene. At 7.5 Ma we identify a possible intermediate state (where data may be a mix of both open and closed cases), with low M-scores from $\delta^{13}C$ in both CAS open and CAS closed "best fit" cases. At 12.5 Ma our combined $\delta^{13}C$-temperature approach showed very poor agreement with other indicators of climate changes, for example in reconciling surface ocean temperature

reconstructions with evidence of extensive cooling and ice growth at this time. Importantly, benthic $\delta^{13}C$ patterns are able to distinguish between open and closed CAS for the present-day in the model, which we found is not the case for benthic temperature proxies.

## 5 Conclusions and summary

In this study, we used proxy data estimates for both surface and benthic temperature ($\delta^{18}O$) as well as benthic $\delta^{13}C$, to constrain

the evolution of atmospheric eq.$CO_2$ and large-scale ocean circulation in the 'cGENIE.muffin' Earth system model and hence identify plausible climatic states in cGENIE for each of 7 time-slices spanning the mid Miocene to Holocene cooling. Constrained by changes in the absolute magnitude and pattern of benthic $\delta^{13}C$, we also diagnosed a plausible history of atmospheric $\delta^{13}CO_2$ over this time interval for use as boundary conditions in future modelling studies, or as a data target for assimilation in geochemical box-models.

In the cGENIE model, we diagnose a progressive reduction in atmospheric greenhouse gas forcing since the mid Miocene, driving global cooling. Simultaneously, we diagnose a gradual strengthening of overturning circulation in the Atlantic that transports heat to the deep Atlantic ocean over the cooling period and leading to a stronger cooling in surface (at ~6°C) than in deep waters (at ~3°C) occurring since the middle Miocene. This onset and strength of the AMOC in cGENIE is controlled by the combined effects of progressive restriction of the Central American Seaway together with our two free

parameters – a salinity adjustment (representing mountain building in N. America and an increasing Atlantic-Pacific salinity gradient), and a declining atmospheric $CO_2$. Declining $CO_2$ drives cooling which helps to promote the sinking of salty waters in the N. Atlantic. The net result in the model is a strong and deep AMOC in the Holocene when the CAS is closed and atmospheric $CO_2$ is low.

**Appendix A. Correcting benthic temperature calculated from δ¹⁸O for local salinity**

The Cramer et al. (2011) $\delta^{18}O_{sw}$ estimate for sea water is a global mean value. Applying the Marchitto et al. (2014) paleotemperature calculation using one global mean $\delta^{18}O_{sw}$ therefore results in an uncertainty in temperature depending on location (Fig. A1) due to local differences in $\delta^{18}O_{sw}$ values (Fig. A2).

The seawater $\delta^{18}O_{sw}$ is determined by global ice volume (which we get from Cramer et al., 2011), local temperature and local salinity (Rohling, 2013). In the present day with an active AMOC, the North Atlantic benthos have a more positive

$\delta^{18}O_{sw}$ than, for example, the North Pacific. This is due to both temperature and salinity, with the salty North Atlantic waters transported to the deep by the AMOC. In our model ensembles the benthic salinity is affected by both $CO_2$ and flux correction as these affect ocean circulation. Therefore, for each simulation we apply a $\delta^{18}O_{sw}$ driven correction to the paleo temperatures due to their local modelled salinity. To do this, we use present day deep-water (2500m) $\delta^{18}O_{sw}$ (LeGrande and Schmidt, 2006) and salinity (WOA 2013, Zweng et al., 2013) to create a general linear model (Eq. A1), where S is salinity.

$$\delta^{18}O_{sw} = 0.8\,S - 27.7 \qquad (A1)$$

The North Atlantic is the region which is most affected by changes in salinity, and therefore the greatest temperature offset will be in this location. We adjusted the salinity model to get best-fit values for the North Atlantic, and a good fit for the difference in $\delta^{18}O_{sw}$ between the North Atlantic and the Pacific. All ocean locations with $\delta^{18}O_{sw}$ between -0.3 and 0.3‰ are shown in a crossplot of data and salinity-model derived $\delta^{18}O_{sw}$ in Fig. A3. The grouping of points clearly offset from the 1:1

line are the high southern latitudes (also clearly visible in Fig. A2). The accuracy of the salinity-derived model in finding $\delta^{18}O_{sw}$ is ±0.03‰ at one standard deviation, and ±0.06‰ at the 95% confidence level excluding latitudes higher than 70° (where we have no formainifera shell $\delta^{18}O$ datapoints in any case).

The paleotemperature equation we apply is the linear model from Marchitto et al. (2014, Eq. 8 therein), using the global ice volume estimate from Cramer et al. (2011). The $\delta^{18}O_{sw}$ model from salinity is also linear so we apply a simple linear

correction to the calculated temperature (in Supplementary Table S2). To obtain the $\delta^{18}O_{sw}$ offset, we first find the modelled global mean salinity (not including latitudes higher than 70°), and subtract this from the modelled benthic salinity, giving a $\Delta S$ field (an offset from the mean). We apply Eq. A2 to find $\Delta T$, where 0.8 is the gradient of the linear $\delta^{18}O_{sw}$ model (Eq. A1) and 0.224 is the gradient of the linear paleotemperature model (Marchitto et al., 2014), and use this to correct T for salinity. The uncertainty for the temperature correction is ±0.13 °C at one standard deviation and 0.23 °C at the 95% confidence level (based

on the $\delta^{18}O$ model uncertainties). The range of temperature correction corresponding to the benthic $\delta^{18}O_{sw}$ spread (of ~0.4‰) in the present day is 1.8°C.

$$\Delta T = \frac{0.8\,\Delta S}{0.224} \qquad (A2)$$

**Appendix B. Model boundary conditions**

In this paper and as per in previous (deeper time) paleo applications of the cGENIE.muffin model (e.g. Ridgwell and Schmidt, 2010), rather than observations, we derive the required boundary conditions from a representative fully coupled GCM experiment using the 'muffingen' open-source software version v0.9.20, which is assigned a DOI: 10.5281/zenodo.4615664. This software takes a specific GCM experiment as input, and carries out the following:

1. Creates output (here: 36×36 with 16 levels in the ocean) grids.

2. Using (1), derives a land-sea mask and also ocean bathymetry on the output grid. The land-sea mask is filtered to prevent the occurrence of isolated in-land seas and single-cell width coastal embayment, while the ocean bathymetry is filtered to avoid single cell 'holes' occurring in the ocean floor.

3. Generates drainage basins determining where precipitation and hence in which direction runoff is directed towards the ocean. The specific scheme used here is known as a 'roofing scheme' and operates to create a watershed approximately equidistant from the coast.

4. Derives island and ocean paths files required by the ocean circulation model.

5. Re-grids GCM wind stress and (10 m) wind velocity to the output grid (which for wind stress, means re-gridding to both u and v edge grids). Wind speed is calculated from the mean annual wind velocity components.

6. Re-grids the GCM planetary albedo and converts to a zonally-averaged profile.

The GCM simulations underlying our cGENIE model configurations were carried out using HadCM3LM2.1aE, which is described in detail in Valdes et al. (2017). The models are constrained with paleogeographies and solar constant appropriate for each geological stage in the Miocene, and a $CO_2$ mixing ratio of 400 ppmv. The experimental design is described in detail in Farnsworth et al. (2019). The specific GCM experiments we used from Farnsworth et al. (2019) are those for: Holocene (0 Ma), Piacenzian (2.58-3.6 Ma), Zanclean (3.6-5.333 Ma), Messinian (5.333-7.246 Ma), Tortonian (7.246-11.63 Ma), Serravillian (11.63-13.82 Ma), Langhian (13.82-15.97 Ma) and for which we simplified and adopted to approximately evenly spaced slices at: 0.0 Ma, 2.5 Ma, 4.5 Ma, 7.5 Ma, 10.0 Ma, 12.5 Ma, and 15.0 Ma (Table 1 main text).

For all time-slice configurations we generally follow the default re-gridding algorithm of the muffingen software in order not to impose prior assumptions regarding the importance of specific ocean features, meaning that for most of the reconstructed continental configurations, a Mediterranean Sea is not present in cGENIE. Only in the 12.5 and 15.0 Ma time-slices is the remnant Tethys Ocean sufficiently expansive to re-grid as an ocean basin at the selected 36×36 (16 levels) model resolution. We do, however, make the following manual interventions in the generation of the land-sea mask (but not in ocean bathymetry):

- For 0 Ma (late Holocene) – the Panama Isthmus as well as the tip of S. America is made continuous.
- For 10.0, 12.5, and 15.0 Ma – the Arctic is opened up (turning land cells to ocean) in order to approximately preserve, across all the 7 reconstructions, the global land fraction in the underlying GCM of 0.29.

- In a second set of simulations the Panama Isthmus is made continuous for time slices 10 Ma to 2.5 Ma inclusive, and the late Holocene Panama Isthmus is left open.

The resulting cGENIE bathymetry for each time-slice is shown in Fig. 3. The Drake Passage is already open at 15 Ma, but widens by 12.5 Ma and gradually deepens. The Atlantic Ocean widens and deepens throughout, and the Pacific generally deepens. The Australian land mass moves north and the Indonesian seaway gradually reduces. Africa moves north, reducing the Tethys Sea area, which is already closed off from the Indian Ocean at 15 Ma. The Tethys disappears by 10 Ma, and the Mediterranean Sea is not included in the simulations. Greenland is isolated from North America until 7.5 Ma, and the Bering Strait is closed until 7.5 Ma. Only at the Holocene is both Bering Strait open and Greenland is isolated from N. America once again, allowing mixing between the Arctic ocean, the North Pacific, and North Atlantic at the same time.

We chose and calculate a zonally average (GCM-derived) planetary albedo profile (rather than a 2D re-gridded one) in order to retain closer back-compatibility with the original GENIE configuration in which an idealised zonal profile is applied (e.g. Edwards and Marsh, 2005). Different GCMs average and save wind speed differently (or only as velocity vectors), meaning that the final re-gridded wind speed product can differ substantially between GCMs and with modern observations. Because of this, we re-scale air-sea gas exchange in the late Holocene configuration in order to give a mean global and annual average modern air-sea coefficient value for $CO_2$ of approximately 0.058 mol m$^{-2}$ yr$^{-1}$ μatm$^{-1}$. This same air-sea gas exchange scaling is then applied to all older time-slices. As compared to Cao et al. (2009), we also forego the high southern latitude zone of reduced atmospheric diffusivity, previously used for present-day model configurations (described further in Marsh et al., 2011). Initial mean salinity was reduced by 1 PSU to 33.9 PSU in all time-slice configurations for simplicity and consistency (although we recognise that in reality mean ocean salinity should progressively decrease from a modern value of 34.9 back in time with progressively decreasing global land ice volume). The orbital configuration was kept at its modern settings throughout all time-slices (0.0167 eccentricity, 0.397789 for sine of obliquity, 102.92 for the longitude of perihelion (in degrees)). However, we did vary the following model parameter values and initial conditions as a function of (geological) time (time-slice age):

- The solar constant is assumed to change with time and to follow Gough (1981) (and see: Feulner, 2012), resulting in a small increase between 15.0 and 0 Ma, from 1366.09 Wm$^{-2}$ (a reduction of 0.14% compared to modern) at 15.0 Ma, to 1368.0 Wm$^{-2}$ by the late Holocene.

- The mean Mg/Ca ratio of the ocean is also assumed to change with time, following Tyrrell and Zeebe (2004). The corresponding range is then from 13.15 mmol kg$^{-1}$ Ca$^{2+}$, 41.21 at mmol kg$^{-1}$ Mg$^{2+}$ at 15.0 Ma, to 10.28, 52.81 mmol kg$^{-1}$ in the present-day ocean. (Changing ocean Mg/Ca with time influences the calculation of carbonate saturation as well as dissociated constants).

All boundary configurations and relevant parameter settings (as well as the cGENIE.muffin model code itself) are open-source. Refer to our 'Model code availability' statement regarding obtaining and running any or all of these model configurations.

## Appendix C. Sustained climate oscillations in the ensemble

The settings in which sustained climate oscillations were identified are shown in Fig. C1. These oscillations were present in a subset of ensemble members in all timeslice ensembles except the Holocene, and were more prevalent for the open CAS configurations. They ranged in amplitude from 0.3°C to 2.5°C in benthic temperature, and periods of 1.7 kyrs to 4.5 kyrs.

## Appendix D. Details on selecting the "best fit" settings

Holocene, 0 Ma

For the Holocene slice, we have a priori knowledge that $CO_2$ at this warm interglacial period was around 280ppm (Indermühle et al., 1999) – a value which in cGENIE agrees with the surface ocean temperature data. This gives us some confidence in the cGENIE Earth system model and the methodology, but caveated by the fact that although we use here a new and different GCM-derived (Farnsworth et al., 2019) modern continental configuration, cGENIE has already previously been calibrated against present day observed ocean temperatures at an atmospheric $CO_2$ value of ca. 280 ppm (e.g. Price et al., 2009; Ridgwell et al., 2007), and so an acceptable fit to SSTs for a eq.$CO_2$ value of 280 ppm is not necessarily unexpected. The global benthic temperature estimate from Cramer et al. (2011) represents a mean climate for both glacial and interglacials, which is around 2 °C cooler than the warmer Holocene. For a global benthic temperature warmer by 2 °C compared to Cramer et al. (2011), a relatively low flux correction of 0.1 to 0.2 Sv is required to fit the data (in comparison, the present-day calibrated value in cGENIE is 0.32 Sv). However, the benthic $\delta^{13}C$ data constraints (Fig. 6) tends towards a higher M-score for a higher flux correction, so we chose a best fit value of FwF 0.2 Sv for the CAS closed case. In a CAS open case, the $\delta^{13}C$ constraint suggests a higher FwF, so the CAS open best fit is set at 0.3 Sv.

Piacenzian, 2.5 Ma

For the 2.5 Ma slice, surface and benthic temperatures both suggests a higher (and larger range for a fit to) eq.$CO_2$ than at Holocene, so 400 ppm is selected as the best-fit eq.$CO_2$ value (although 280ppm would also work well). Similar as for the Holocene, a higher flux correction (FwF) at 2.5 Ma tends to show a better model-data benthic $\delta^{13}C$ agreement for the CAS open case, although this constraint is less strong than the benthic temperature constraint (maximum M-score for benthic temperature is ~0.4, and for $\delta^{13}C$ at 0.5Sv is ~0.2). We hence select 0.3 Sv as the FwF value, on the higher end of the benthic temperature ($\delta^{18}O$ derived) dataset, and in agreement with global mean benthic temperature from Cramer et al. (2011) for the open CAS. The closed CAS shows a generally cooler benthic ocean temperature, and a lower FwF, so a eq.$CO_2$ of 400ppm is combined with FwF of 0.1 Sv for the closed CAS.

Zanclean, 4.5 Ma

At 4.5 Ma, the surface temperature data supports a higher again eq.$CO_2$ value in cGENIE, but here the $\delta^{13}C$ constraint is stronger (as compared to that at 2.5 Ma) for both CAS cases, and instead supports a lower-$CO_2$-higher-FwF, as the

M-score increases in this direction of parameter space (Fig. 6). We hence place the best-fit in this direction for CAS-open, on the higher end for the benthic temperature constraint range, and with eq.$CO_2$ at 400 ppm and a FwF of 0.5 Sv. For the closed CAS the benthic $\delta^{13}C$ constraint is stronger still, and, together with a benthic temperature constraint indicating a lower FwF than for the open CAS, an eq.$CO_2$ of 400ppm is combined with FwF of 0.2 Sv for the closed CAS.

Messinian, 7.5 Ma

At 7.5 Ma, surface-temperature data requires requires a significantly increased eq.$CO_2$ compared to 4.5Ma, with 800 ppm as being clearly the best-fit $CO_2$ value. A similar tendency (to 4.5 Ma) to lower $CO_2$ but higher FwF is also apparent in the $\delta^{13}C$ constraint for this time-slice, although at 800pm this is less strong (with a low M-score for the $\delta^{13}C$ constraint at 800 ppm and high FwF). Again, with a closed CAS the benthic temperature constraints indicate lower FwF than for the open CAS. The flux correction is set at 0.4 Sv, near the centre of the benthic temperature maximum M-score (shown as a dashed blue contour) for the open CAS case. For the closed CAS, 800ppm is combined with FwF of 0.3 Sv, towards the higher end of the benthic temperature constraint, but in the direction of increasing $\delta^{13}C$ M-score.

Tortonian, 10 Ma

For 10 Ma and older time slices the model-data M-score for surface temperature declines, but with higher scores for higher $CO_2$ in all three cases. At 10 Ma, for the open CAS, the benthic ($\delta^{18}O$) temperature dataset suggests a $CO_2$ of 800 ppm and FwF of 0.3 to 0.4 Sv. The $\delta^{13}C$ fit surface tends towards higher M-score values for higher flux, so we set FwF at 0.4 Sv for CAS open. For CAS closed, both benthic constraints suggest a lower FwF, so 800ppm combined with FwF of 0.3 Sv is selected for CAS closed.

Serravalian, 12.5 Ma

The 12.5 Ma slice seems to show a transition state for the trends in $\delta^{13}C$, with overall low M-score for all combinations of $CO_2$ and FwF. As $\delta^{13}C$ provides a weak constraint, the $CO_2$ and FwF values are selected as 1120 ppm and 0.2 Sv, respectively, as a compromise between the high $CO_2$ requirement for surface temperature and the lower-$CO_2$ higher-FwF for the benthic temperature.

Langhian, 15 Ma

At 15 Ma the ensemble surface of M-score for $\delta^{13}C$ is inversed compared to other timeslices, showing a higher score for (generally) higher eq.$CO_2$ combined with lower FwF. Although we have fewer surface temperature data-points, they suggest a high eq.$CO_2$, somewhat in disagreement with the benthic temperature dataset that favours a mid-range eq.$CO_2$ together with higher FwF. As the $\delta^{13}C$ constraint tends towards a lower FwF (compared to time slices younger than 10 Ma), we select a eq.$CO_2$ of 1120ppm and a low flux correction of 0.1 Sv.

**Supplementary information**

Data tables of: surface ocean temperature data, benthic ocean $\delta^{18}O$ data and calculated non-salinity-corrected benthic temperature~~T~~, benthic ocean $\delta^{13}C$ data.

Time-slice maps showing location of benthic data datapoints and the ±1 million year timeseries for each.

**Model code availability**

The code for the version of the 'muffin' release of the cGENIE Earth system model used in this paper, is tagged as v0.9.22, and is assigned a DOI: 10.5281/zenodo.4741336.

Configuration files for the specific experiments presented in the paper can be found in the directory: genie-userconfigs/MS/crichtonetal.CP.2021. Details of the experiments, plus the command line needed to run each one, are given in the readme.txt file in that directory. All other configuration files and boundary conditions are provided as part of the code release.

A manual detailing code installation, basic model configuration, tutorials covering various aspects of model configuration and experimental design, plus results output and processing, is assigned a DOI: 10.5281/zenodo.4615662.

**Author contribution**

KAC collated the temperature, $\delta^{18}O$ and $\delta^{13}C$ data, devised the salinity correction for temperature and carried out the model-data comparison and analyses; AR produced the cGENIE time-slice configurations, ran the ensembles and extracted mean-climate data; DJL and AF provided the HadCM3L configurations; all authors wrote the manuscript.

**Competing Interests**

All authors declare no competing interests.

**Acknowledgments**

KAC was supported by Natural Environment Research Council (NERC) grant number NE/N001621/1 to PNP. AR acknowledges additional support from the National Science Foundation under Grants 1702913 and 1736771. DJL and AF thank NERC grant NE/K014757/1.

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

**Figures and Tables**

| Data slice (Ma) | HadCM3 slice | | |
|---|---|---|---|
| | Age name | Start (Ma) | Finish (Ma) |
| 15 | Langhian | 15.97 | 13.82 |
| 12.5 | Serravalian | 13.82 | 11.63 |
| 10 | Tortonian | 11.63 | 7.246 |
| 7.5 | Messinian | 7.246 | 5.333 |
| 4.5 | Zanclean | 5.333 | 3.6 |
| 2.5 | Piacenzian | 3.6 | 2.58 |
| Core top | Holocene | 0.006 | |

**Table 1, Dataslices for the carbon cycling approximate locations (in time) and the HadCM3L setups used to create Genie bathymetry. 7.5Ma and the 2.5Ma timeslice are just outside the ages to which they are assigned, but in order to not have two points in the other**
**ages, they place in the Messinian and Piacenzian respectively**

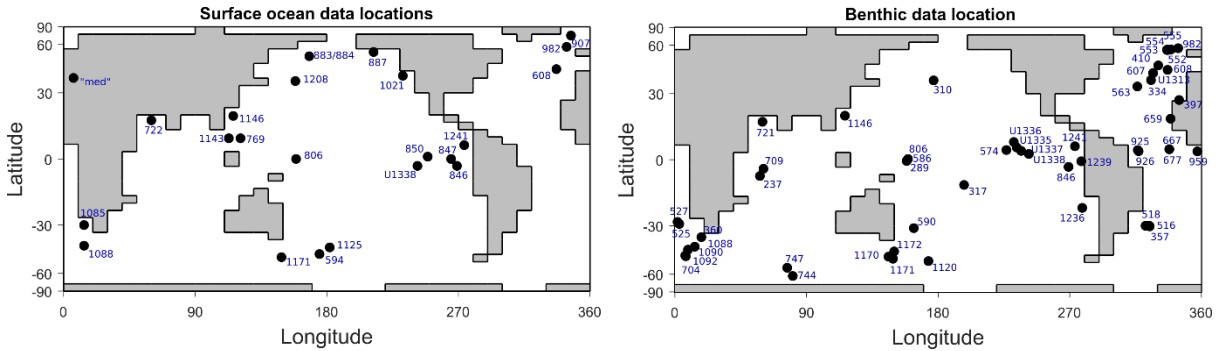

**Figure 1 Present day location of surface ocean temperature data points (left) and benthic δ¹⁸O and δ¹³C data points (right) collated for this study (the Holocene δ¹³C dataset from Peterson et al., 2014 is not plotted).**

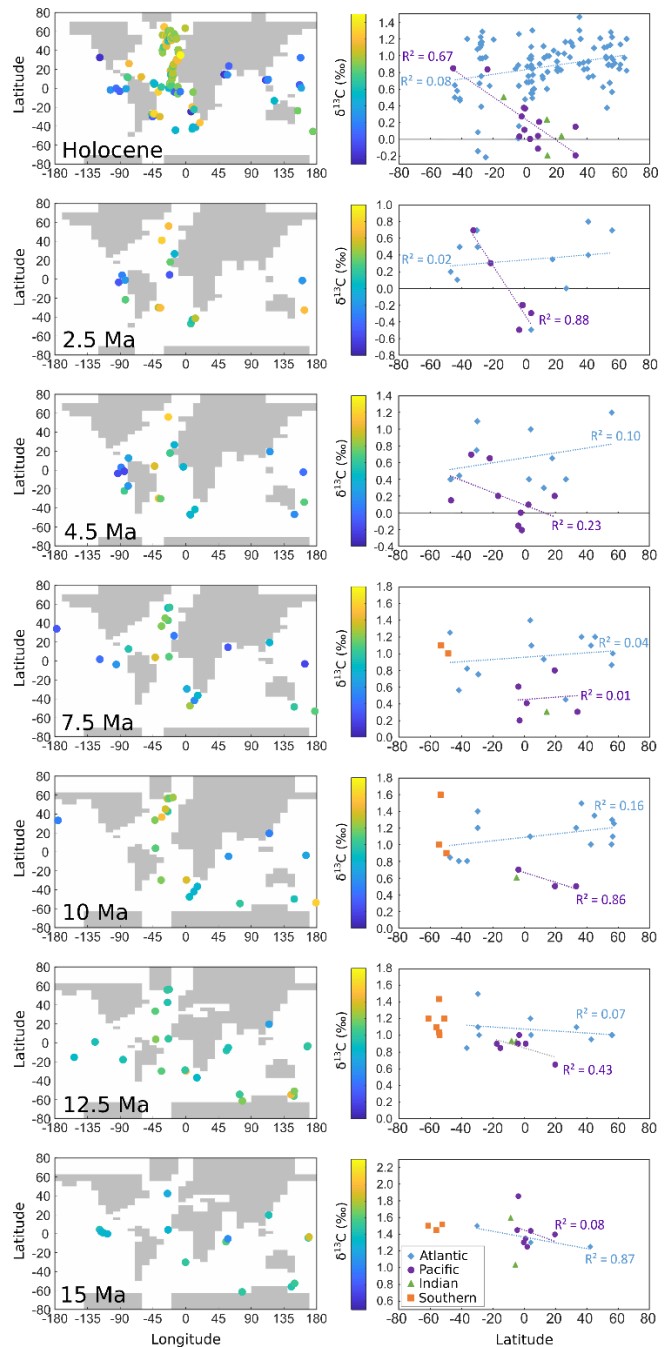

**Figure 2, Benthic δ¹³C data and paleo locations overlaid on paleo-coastlines of HadCM3 regridded to cGENIE scale (left column). Data benthic δ¹³C plotted by latitude for each timeslice, with linear regression and R² for the Atlantic and Pacific basins (right column). The colourscale for the map is also the y-axis scale for the per-latitude data, and all have a range of 1.8‰.**

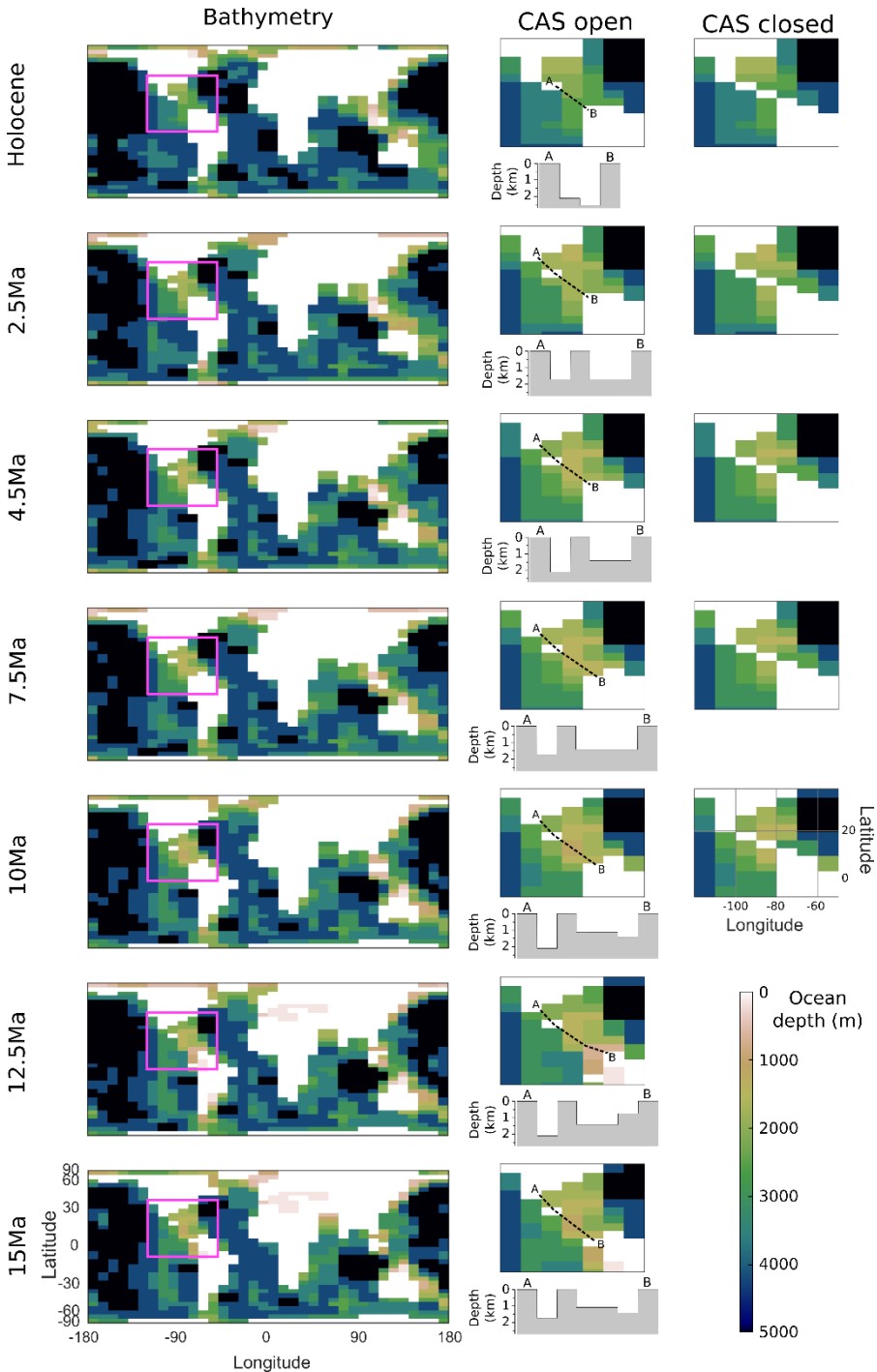

**Figure 3, Model bathymetry and coastlines for each time-slice, and a zoom-in on the region covering the Central American Seaway (CAS) including a transect showing it most depth-restricted pathway for the open-CAS configurations, and the ocean depths for both open and closed CAS configurations.**

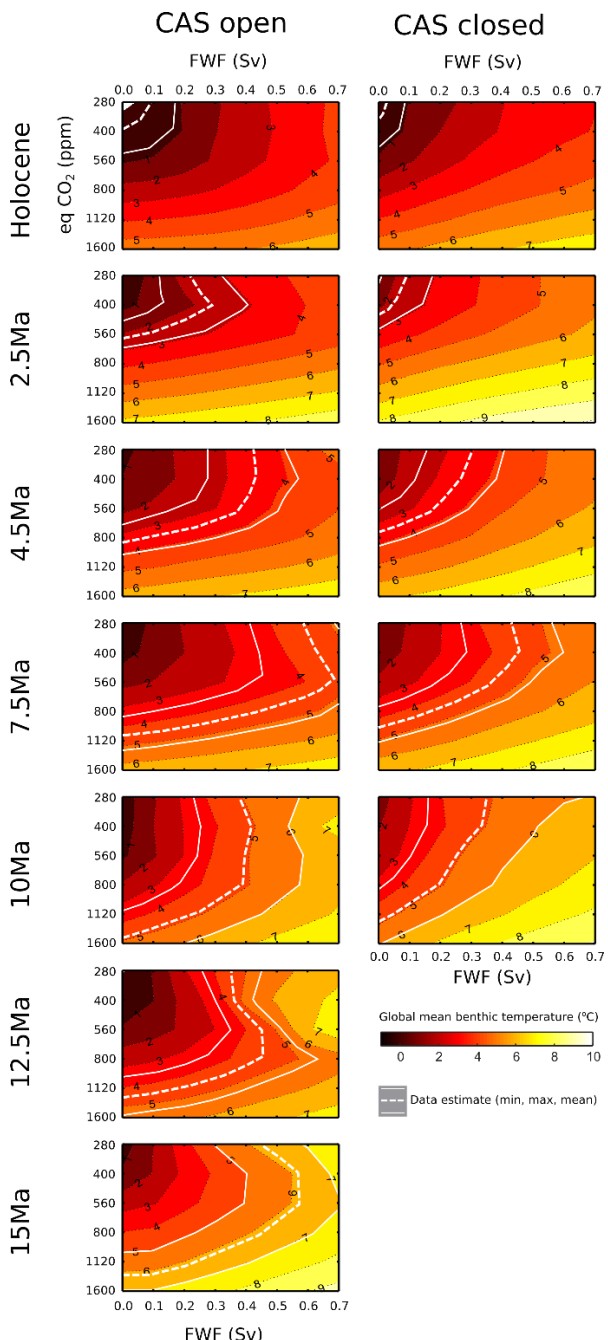

**Figure 4, Modelled benthic temperature with data-global mean temperature estimate (and min max range) (Cramer et al., 2011) overlaid (dashed line and solid white lines respectively). The data-global mean temperature shown for the Holocene timeslice represents 0.1Ma, representing a mean value for Pleistocene glacial-interglacial climate.**

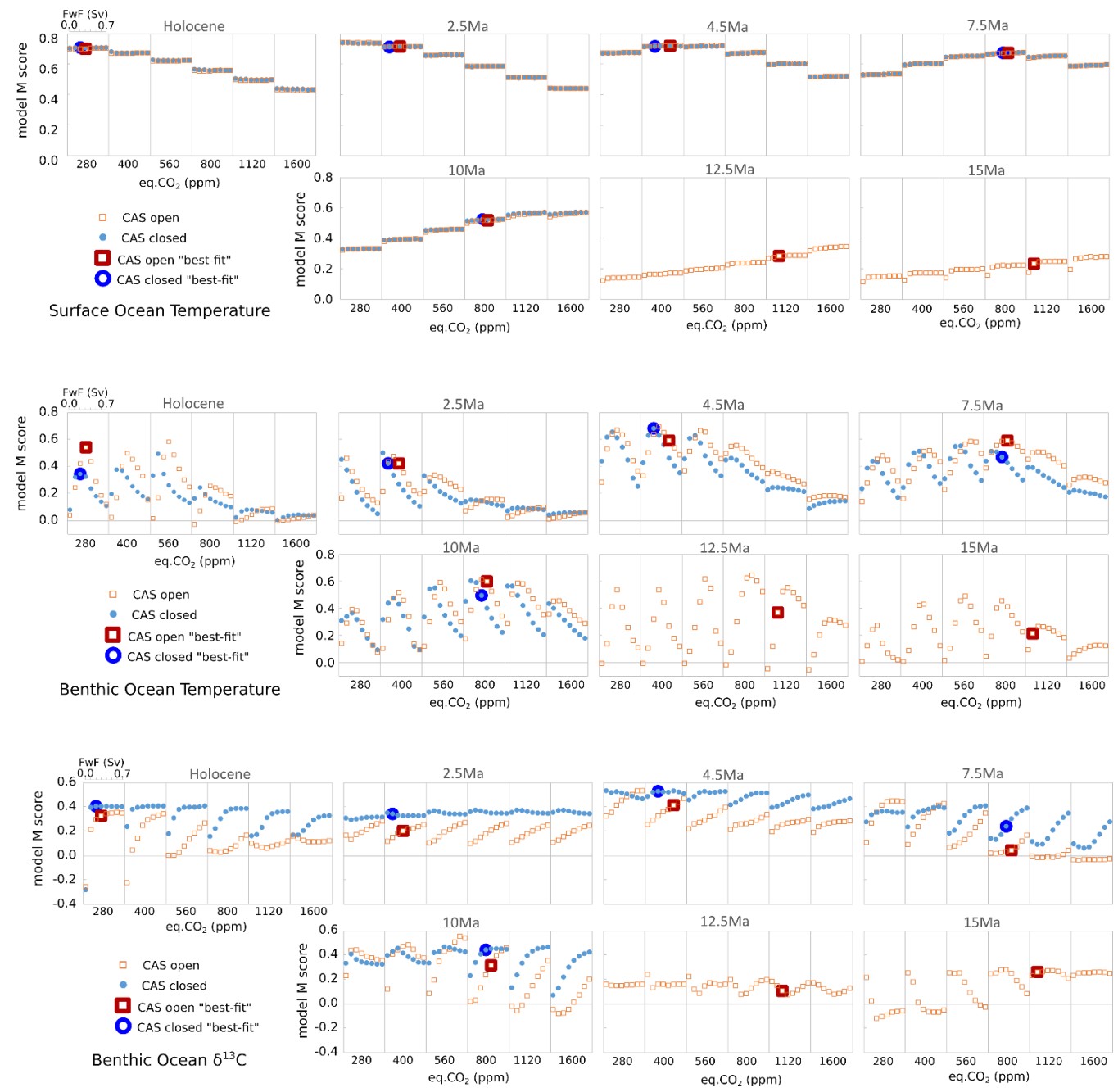


**Figure 5, M-score for fit of model to data for surface temperature, benthic temperature and benthic δ¹³C. Model settings are plotted on the y-axis (totalling 48 simulations per chart).**

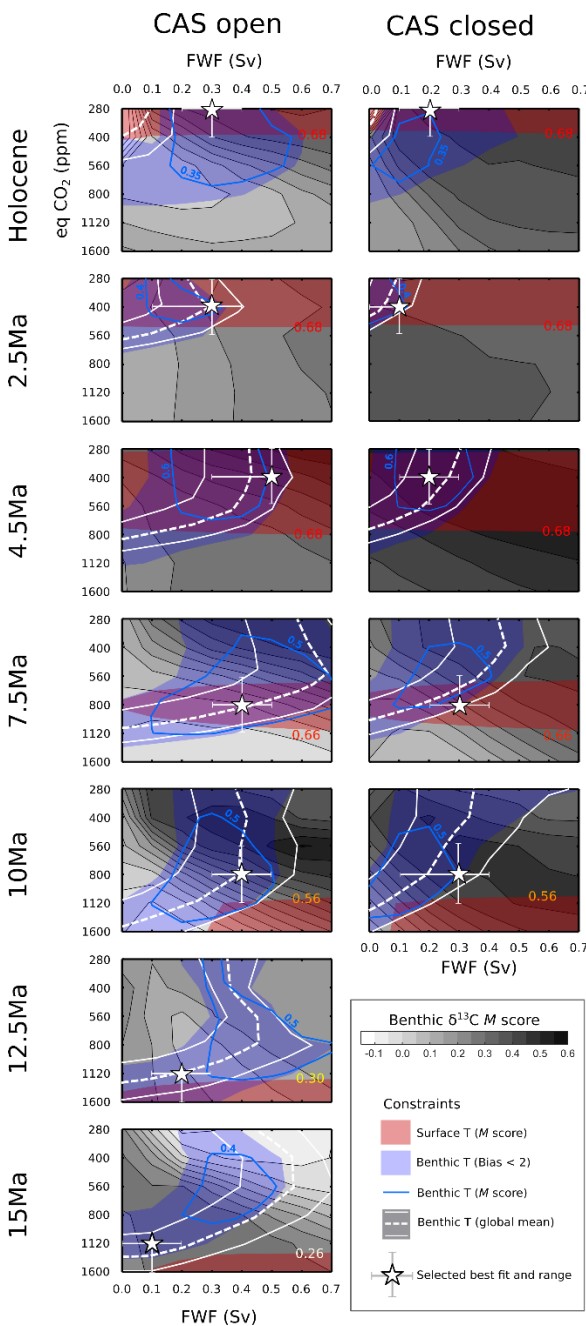

**Figure 6, All proxy data constraints combined.** M-score for modelled benthic δ¹³C fit to dataset (filled contours); modelled benthic global mean temperatures lowest offset to Cramer et al., 2011 (dashed white line, and max min estimates as solid thin white lines); modelled benthic temperature fit to dataset calculated from δ¹⁸O data (blue filled region where mean bias is < 2 °C), blue contour is

**M-score indicator with value marked in blue), for Holocene benthic temperatureT dataset is from WOA 2009; M-score contour for modelled surface ocean temperature fit to data (red shaded section with M-score marked on edge of the contour). Overall best-fit settings (white star, with upper and lower range estimates)**


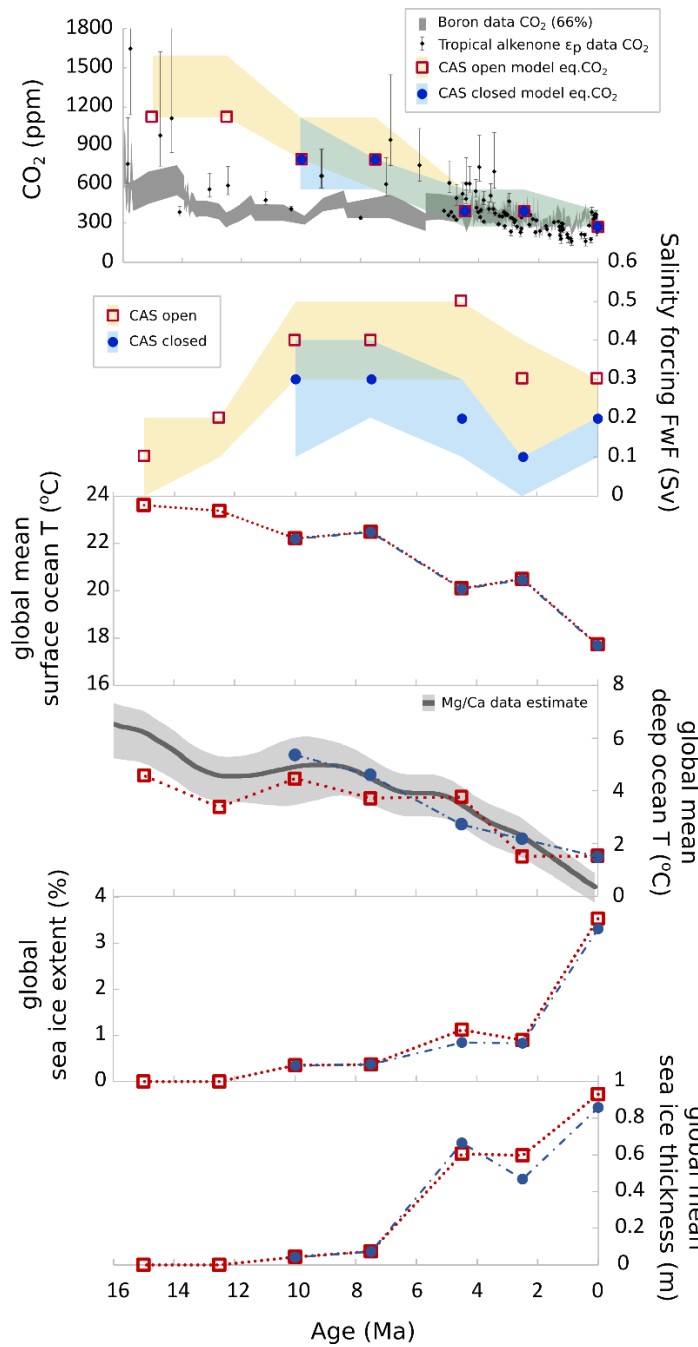

**Figure 7,** The eq.CO₂ and FwF settings (and estimated ranges) that agree with temperature and benthic δ¹³C data constraints according to this study. Plotted with eq.CO₂ are estimates for atmospheric CO₂ levels from two recent studies, tropical alkenones (Stoll et al., 2019) and boron derived (Sosdian et al., 2018). Also shown are: modelled global mean surface ocean temperature, global mean benthic ocean temperature (and the Mg/Ca thermometer estimate from Cramer et al 2011), modelled global sea ice extent, and modelled global mean sea ice thickness for the best-fit settings.

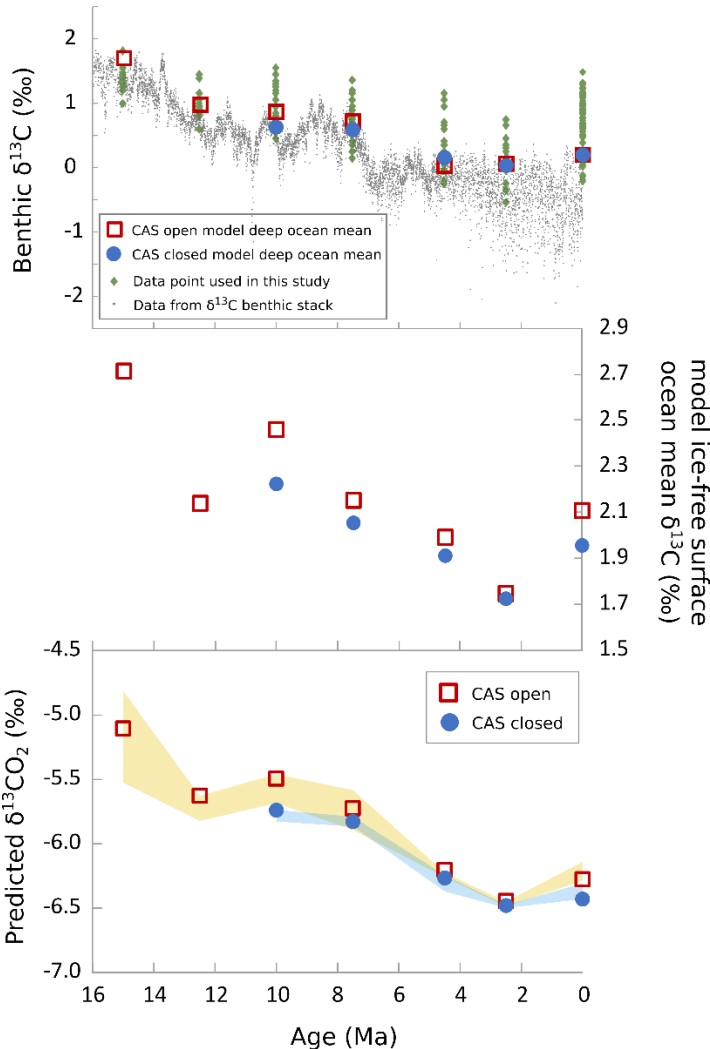

**Figure 8,** Inferred global benthic mean δ¹³C, global mean surface ocean δ¹³C, and atmospheric δ¹³CO₂ calculated using the mean-bias of the best-fit model to the data indicated benthic ocean δ¹³C in each timeslice. The datapoints from the δ¹³C benthic stack are from Westerhold et al. 2020. The shaded regions in predicted δ¹³CO₂ show the range of δ¹³CO₂ using the best ranges identified in Fig. 6.

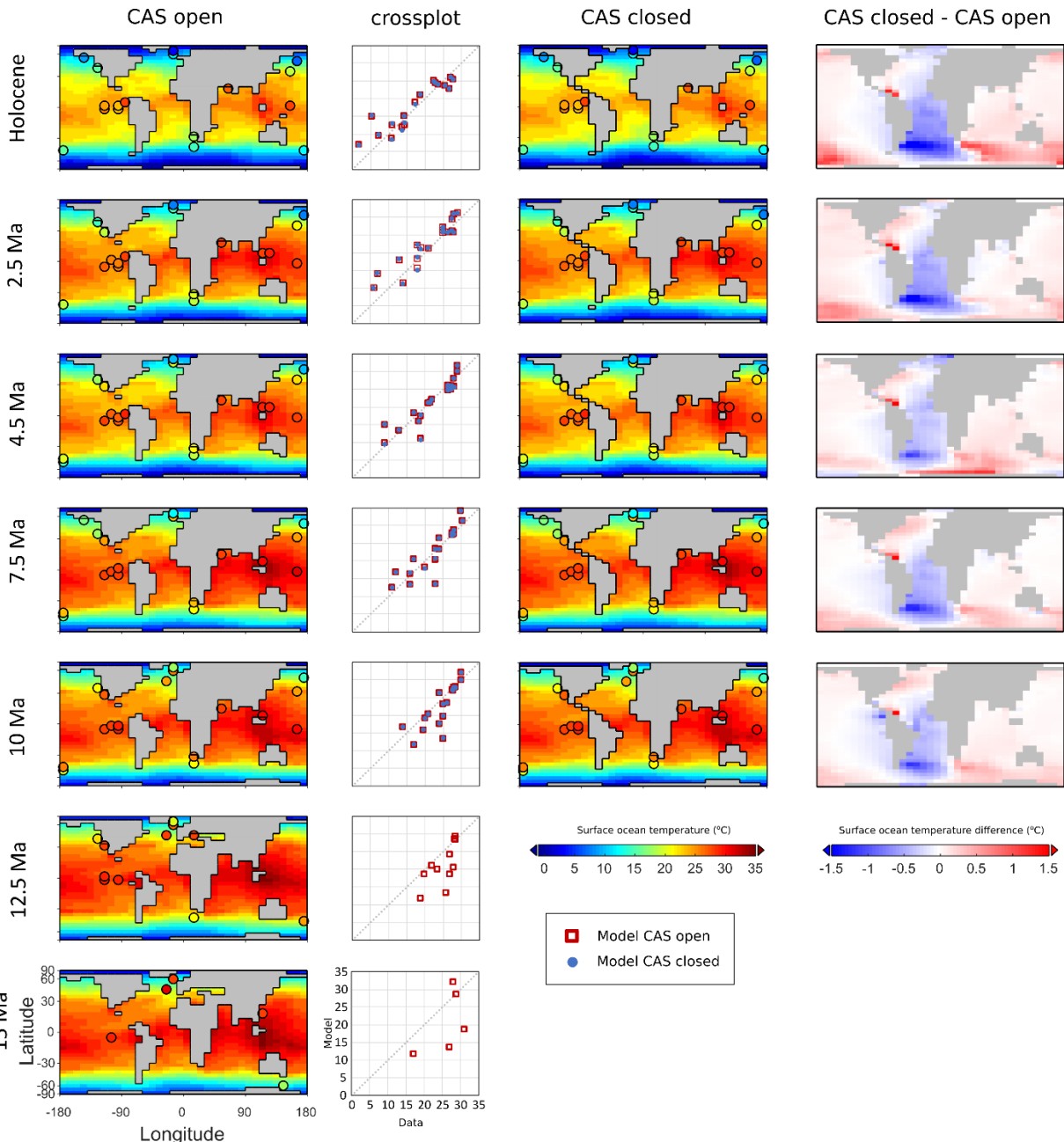


**Figure 9, Modelled and data-indicated surface ocean temperature for best fit model settings with data overlaid as shaded circles (temperature maps); cross plots of data points (x axis) and model points (y axis) for both CAS cases; surface temperature difference shown on the right. The model-data fit statistic (M-score) is shown in fig. 5.**


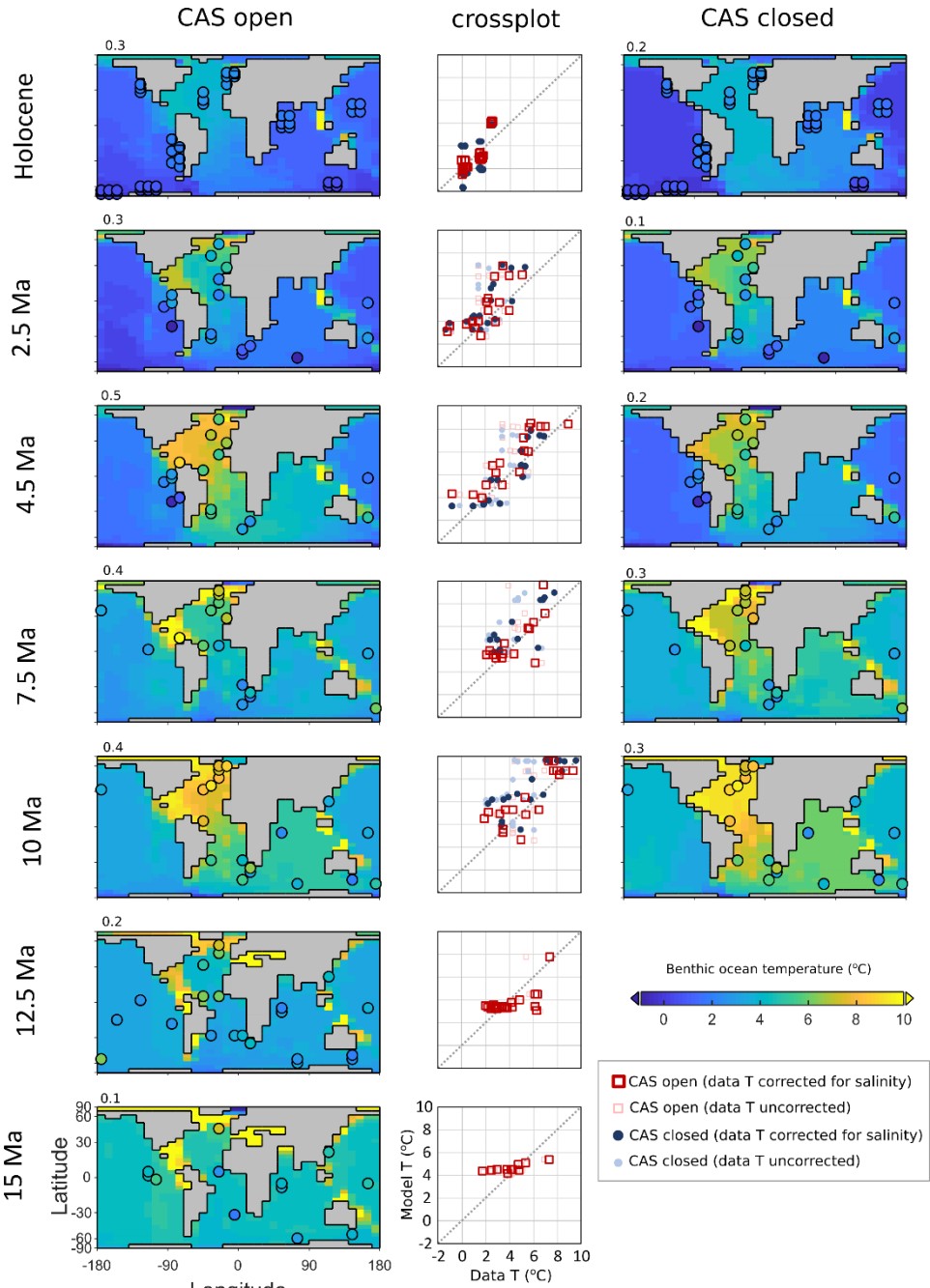

**Figure 10, Benthic temperature for best fit model settings with data overlaid (shaded circles) and cross plots of data points (x axis) and model points (y axis), with temperature calculated with no salinity correction (semi-transparent) and with salinity correction (opaque). Noted above each map is the FwF setting in Sv. ~~The model-data fit statistic (M-score) is shown in fig. 5.~~**

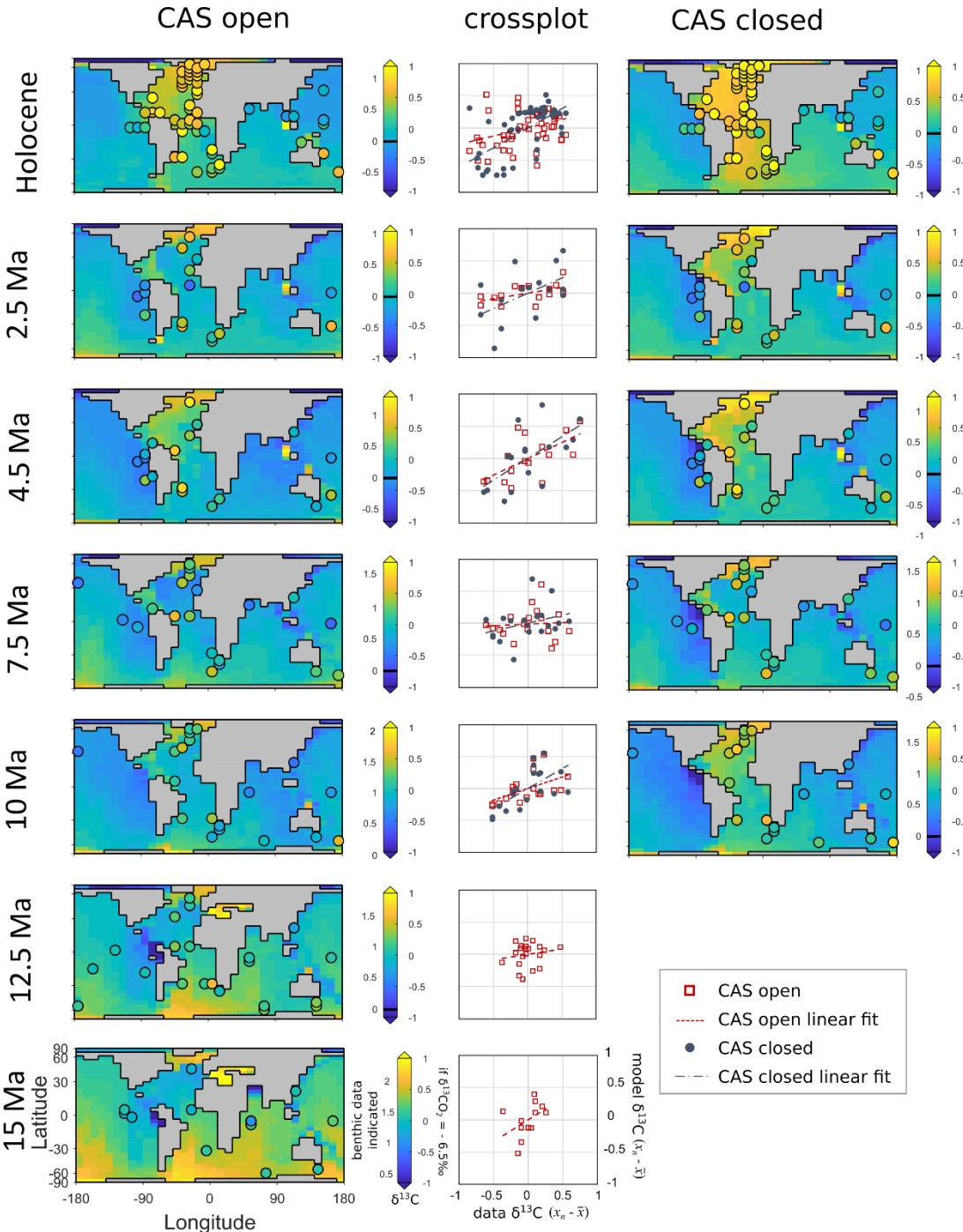

Figure 11, Modelled δ¹³C of DIC for best-fit model settings with data overlaid (shaded circles), and cross plots of mean-adjusted data points (x axis) and mean-adjusted model points (y axis). N.B. Shading is with respect to an atmospheric forcing of δ¹³CO₂ of - 6.5 ‰. This shading allows the comparison of the benthic ocean δ¹³C patterns between the timeslices. The data indicated δ¹³C is shown to the left of each colourbar. **The model-data fit statistic (M-score) is shown in fig. 5.**

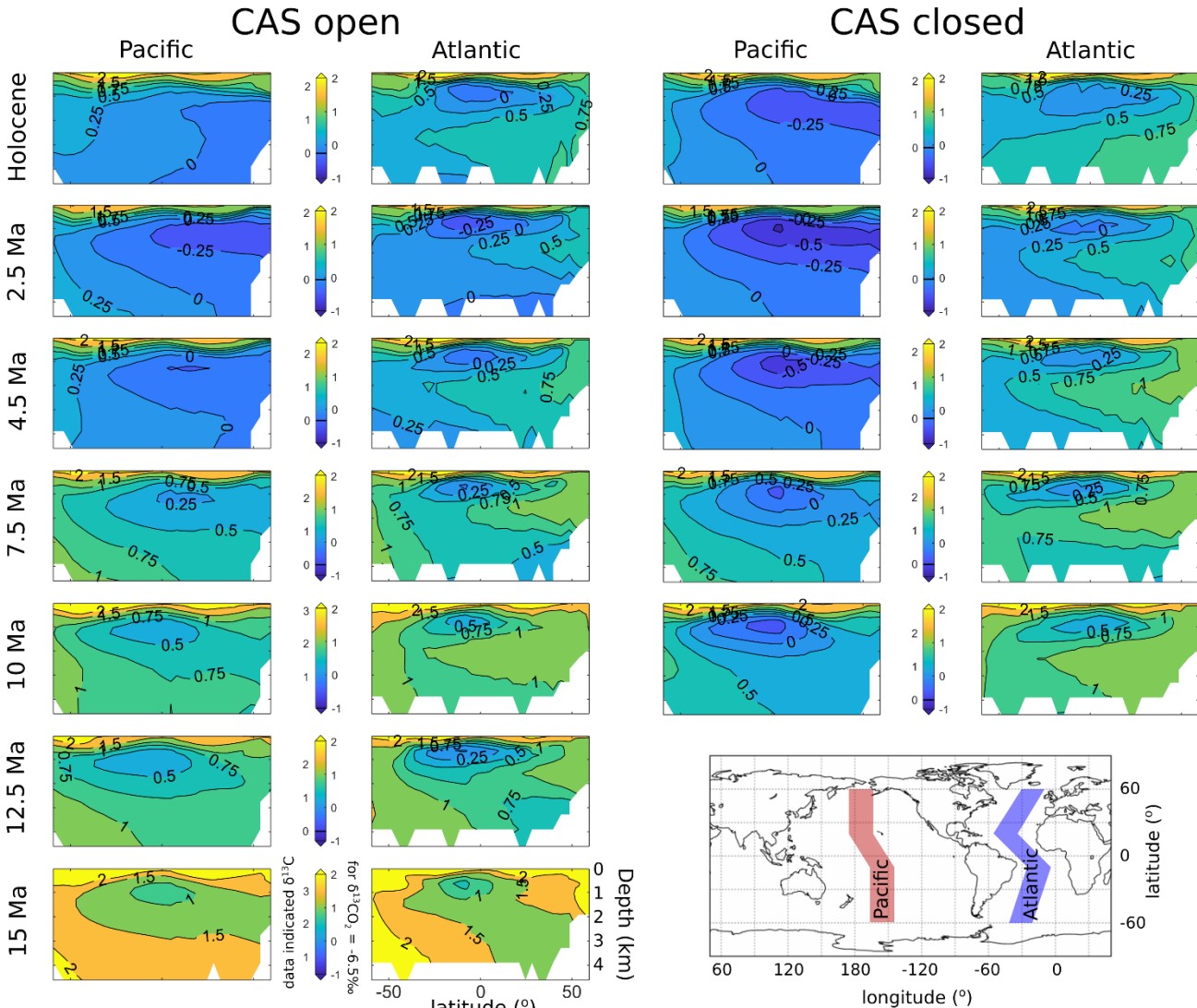

**Figure 12, Pacific and Atlantic ocean cross sections of δ¹³C, shading and contour labelling is with respect to an atmospheric forcing of δ¹³CO₂ of -6.5‰. This shading allows to compare the inner ocean δ¹³C between the timeslices, for example 15 Ma shows values further from the atmospheric value than the Holocene. The data indicated δ¹³C value is shown on the left of each colourbar. The inset map shows the transect locations used to make the cross-sections.**

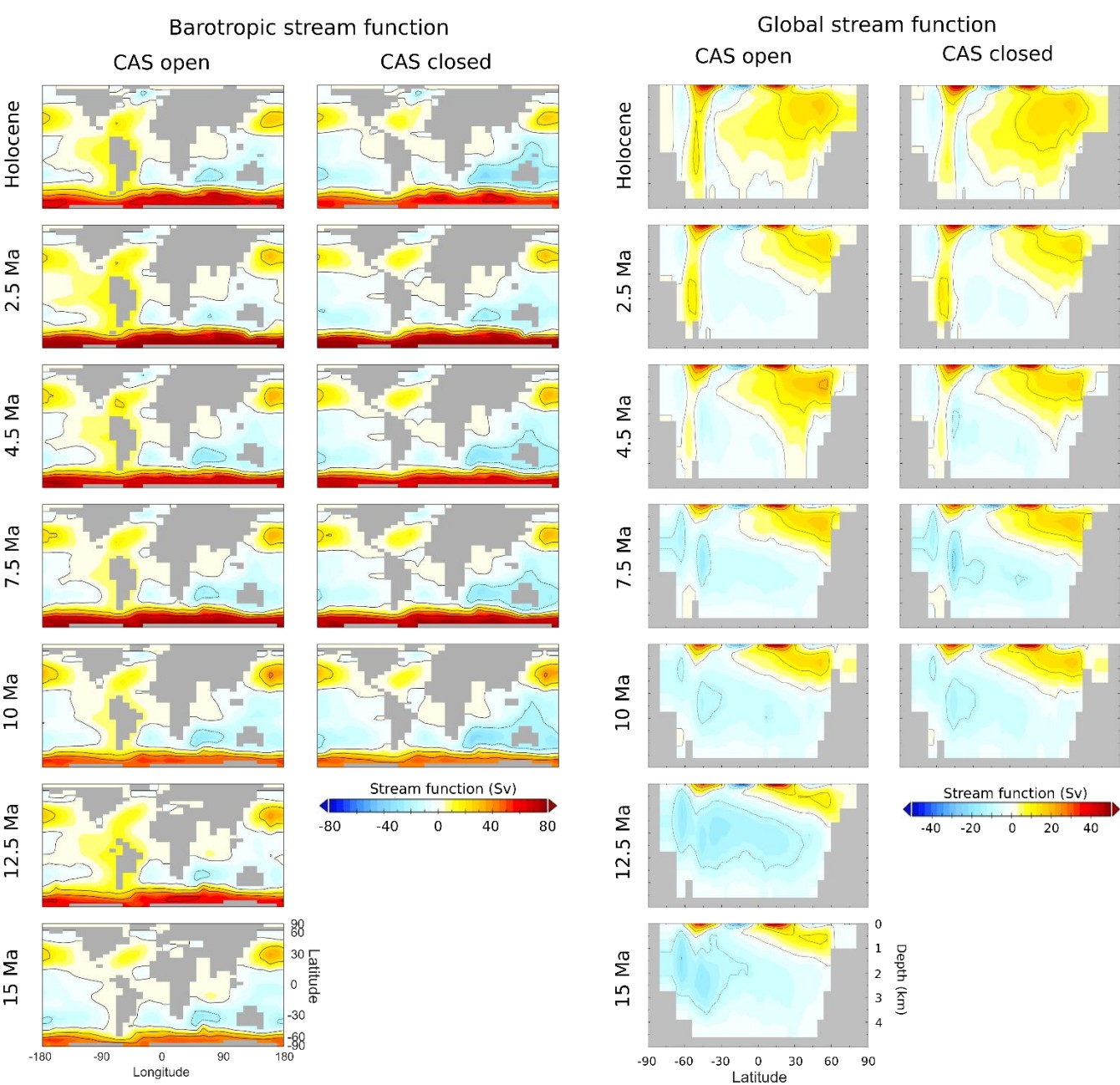

**Figure 13, Modelled barotropic stream function (left) and global stream function (right) for each time slice for the best-fit model settings. Contours for barotropic streamfunction are at 20 Sv intervals, for global stream function at 10 Sv intervals. Note the difference in colourscale between barotropic and global stream functions.**

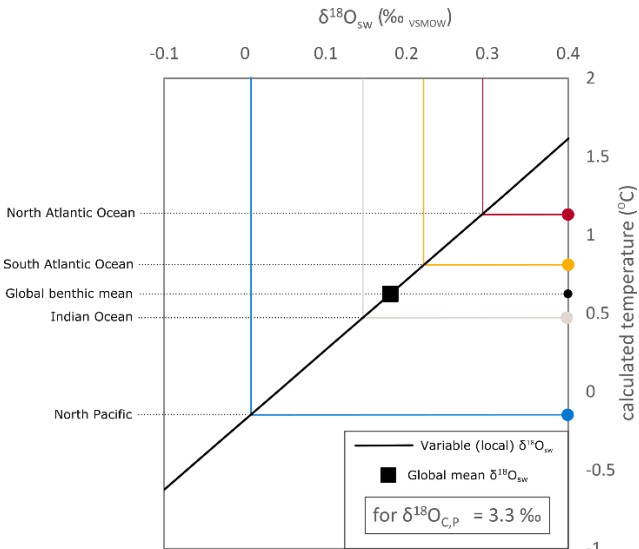

**Figure A1, Example paleotemperature calculation of Marchitto et al. (2014) for a $\delta^{18}O_{Cibicidoides,Planulina}$ of 3.3‰, for a variable $\delta^{18}O_{sw}$.**
**The calculated temperature using the global mean $\delta^{18}O_{sw}$ from Cramer et al. (2011) for the Holocene is marked, and selected ocean region mean $\delta^{18}O_{sw}$ have been labelled on the figure (values evident in Fig. A2 for ocean basins' means). When applying a local correction to $\delta^{18}O_{sw}$, the N. Atlantic calculated temperature is ~0.5 °C warmer and the N.Pacific ~0.7 °C cooler than the when applying the global mean $\delta^{18}O_{sw}$.**

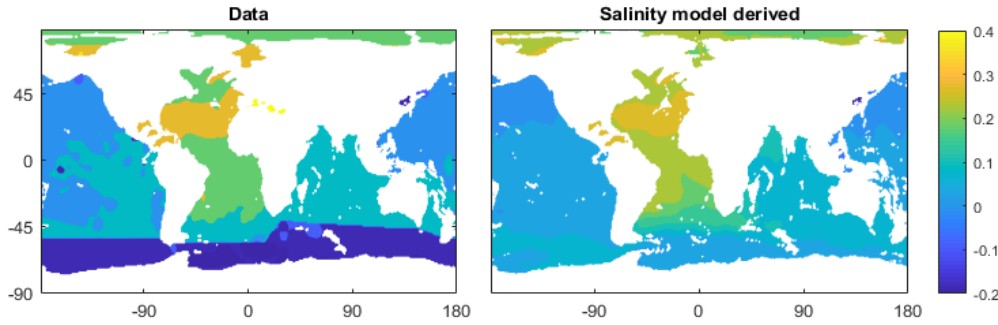

**Figure A2, $\delta^{18}O_{sw}$ data from LeGrande and Schmidt (2006) at 2500m depth (deeper than 2500m the dataset shows little change in $\delta^{18}O_{sw}$ compared to these values shown) (left) and salinity derived $\delta^{18}O_{sw}$ (right) where salinity data from WOA 2013, (Zweng et al., 2013) is used in the model in Eq. A1 (all in ‰$_{VSMOW}$).**

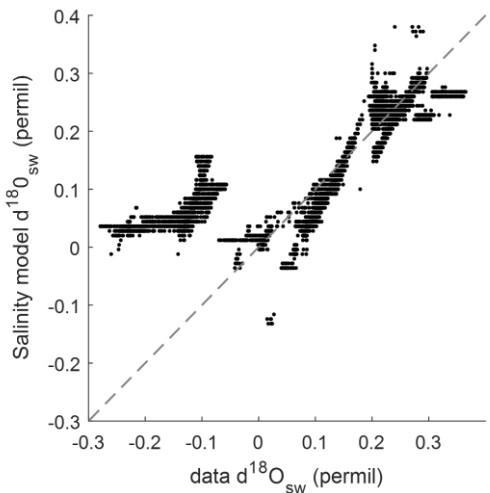

**Figure A3, Cross plot of data δ¹⁸O$_{sw}$ and salinity model derived δ¹⁸O$_{sw}$ as shown in Fig. A2. The offset region (where data shows low δ¹⁸O$_{sw}$ compared to the salinity model derived) is the Southern Ocean, where the polar front more strongly affects ocean water δ¹⁸O$_{sw}$ than salinity/thermohaline circulation.**

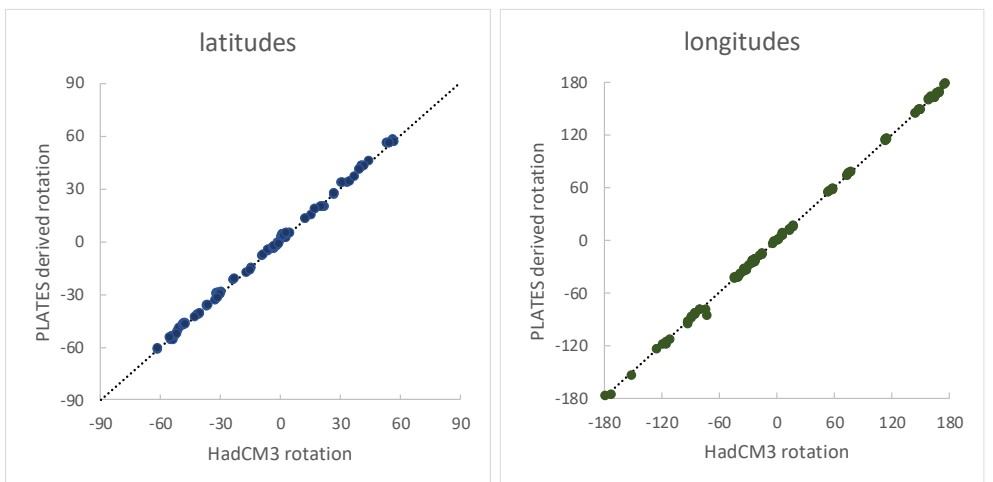

**Figure B1, Comparison of PLATES derived rotation (applied to the full dataset) to the HadCM3 rotation (used as a basis for the cGENIE model). The mean difference between PLATES and HadCM3 paleo-locations for latitude is 0.216° (and standard deviation of 1.14°), for longitude is -0.498° (1.80°).**

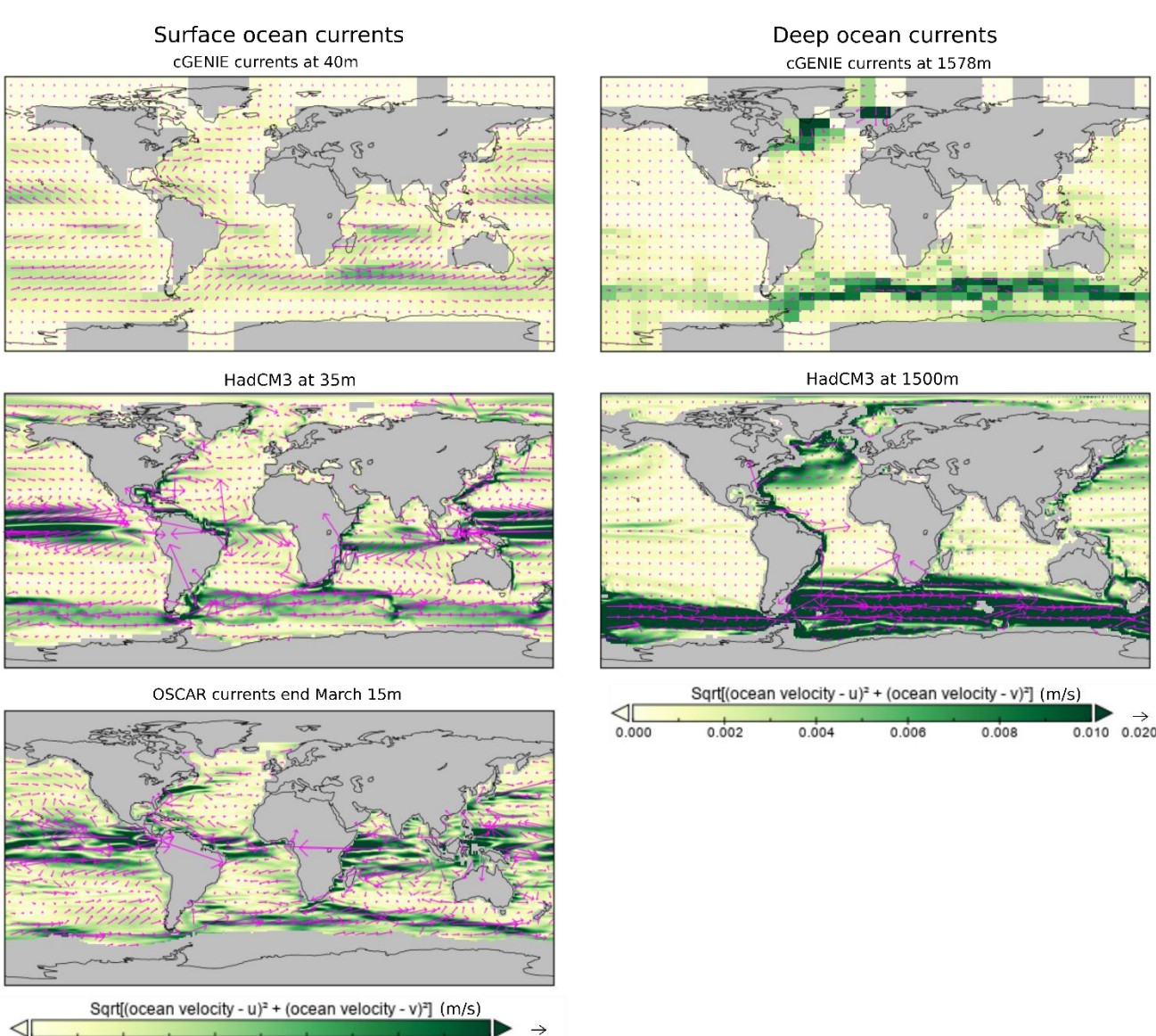

**Figure B2, comparison of cGENIE to HadCM3 near surface and deep ocean currents, and to data-indicated near surface currents from OSCAR (Ocean Surface Current Analysis in Real-time) (ESR 2009). Note the scale difference for the surface and deep colour bars and vectors.**


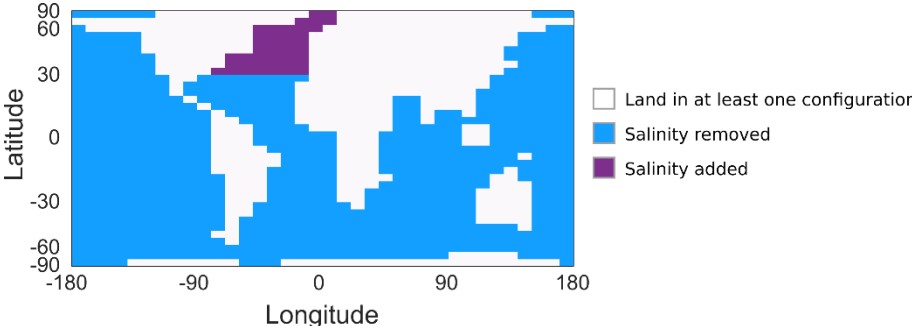

**Figure B3, location of salinity addition and salinity removal common for all configurations for the salinity flux adjustment controller 'FwF'.**

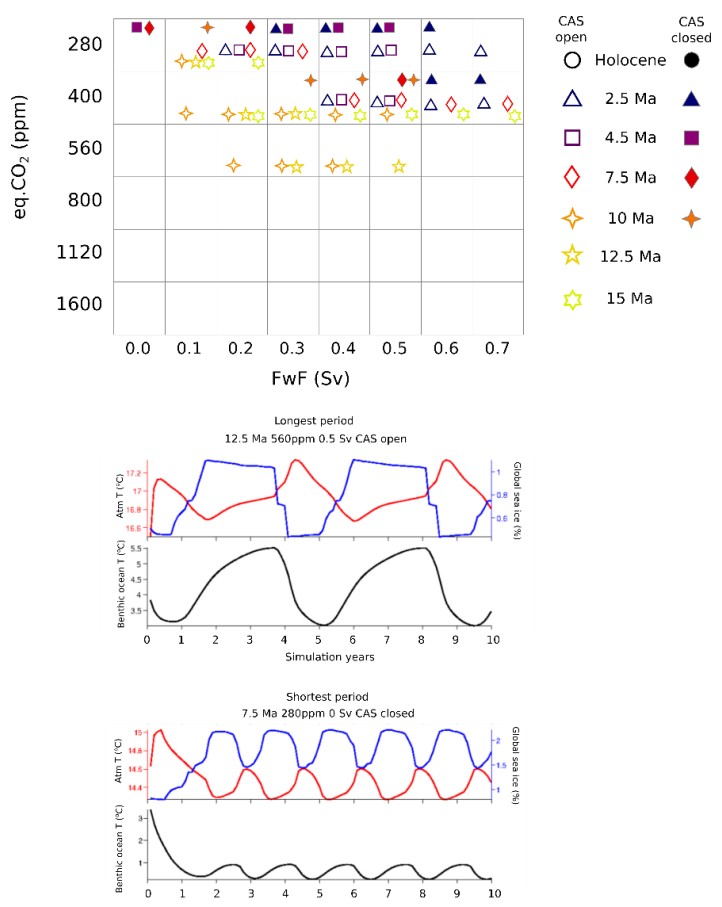

**Figure C1, Ensemble settings that displayed sustained oscillations in benthic ocean temperature (top). Model output for atmospheric temperature, sea ice and benthic temperature for the longest and shortest period oscillations are plotted underneath.**