# Peer review of "Data-constrained assessment of ocean circulation changes since the middle Miocene in an Earth system model"

_Climate of the Past, 2019_

## Referee Comment (RC1) · Anonymous Referee #1 · 17 Feb 2020

General comments - overall quality of the discussion paper

In this paper the authors provides simulations for 7 time-slices from 15Ma to the Holocene using the intermediate complexity model cGENIE, which includes carbon and oxygen cycling. For each time-slice they run several simulations varying 2 parameters they consider to be uncertain/unknown – ie. equivalent $CO_2$ forcing and the magnitude of N. Pacific to N. Atlantic salinity flux adjustment. They then select the combination of parameters that enable the model results to best-fit the data (sea surface temperature from Alkenones and TEX86, benthic $\delta$ 18O and benthic $\delta$ 13C) for each time-slice. In doing so they try to explain the changes seen in the proxy record throughout the

Miocene in term of greenhouse gases forcing and AMOC intensity changes.

This paper provide interesting methodology that can be use to track which process may be lacking in some models to be able to better simulate the climate of past warm periods. Also, we definitely need more 'systematic' methods to test model against data and I do think ensemble set of simulations is a good point to start with.

However I think that the authors should re-organize and re-write the results and discussion part so the main results from this paper are better highlighted. This would be very helpful to help the readers to better understand the purpose of the paper. I list some points below that in my opinion should be taken into account before publication.

Specific comments - individual scientific questions/issues

First, I am a bit concerned regarding some CAS-related statement in the text :

- Wide part of the Discussion focuses on the potential effect of the CAS configuration on circulation proxy, which thus seems to be (one of ?) the main point of the paper. As most of the discussion is CAS-oriented, authors should probably provide a few lines about CAS configuration changes during the Miocene and effect on the global circulation in the introduction using existing literature (e.g. Schneider and Schmittner, 2006 ; Butzin et al. 2011 ; Sepulchre et al. 2014 . . .). From the actual content of the discussion and conclusion it seems that it should also appears somewhere in the abstract.

As authors refers a lot to CAS closure, they should also probably provide as supplementary information a table that contains at least the mean depth of CAS in the different settings they used.

- P17. L.525 " We find that an early CAS restriction probably fits $\delta$13C data better " – It does not seems obvious, while reading the paper that this finding arise from analyzing author's model-data comparison. I also have the feeling that the CAS configurations used in the simulations does not enable to conclude such a thing, as CAS remain very

deep in all the time slices (except for the Holocene). For the most recent cases this is much deeper than what we can find in the literature.

The paper would obviously benefit from additional sensitivity tests on CAS closure, especially for younger time slices (e.g. 2.5 and 4.5 Ma), as it seems, from what the authors states, that this is the missing point to be able to improve model-data fit. It would enable then to conclude whether or not CAS restriction better fits $\delta$ 13C data for the youngest time-slices. The author should also in any case better discuss the relation between CAS configuration and $\delta$ 13C in the light of results from previous studies.

- P16, l. 493 "The flux correction in the N. Atlantic that we apply may be seen as a combination of compensating for a more restricted (or closed) CAS" – The reason why is not very clear at that point of the text. I understand that stating that the authors refers to the fact that closing CAS increase AMOC intensity – or at least change the NADW properties in most models but then authors should refer to those papers and provide a little bit more explanation.

My second issue is with the Results and Discussion parts that are sometimes a bit messy and do not always reflect the purpose of the paper and the main conclusions. I thus suggest that they should be re-written to better emphasize the main results related to change in oceanic circulation, and discussing uncertainties/caveat of the study framework/applied methodology.

- As the title of the paper relates to circulation, I do think that it would be better to start the discussion with circulation-related findings. I feel like the CO2-related issues should come later, introducing caveats of the methodology and model or/and data related uncertainties.

- As the section 4.1 is written now, it looks more like a list of previous Miocene studies than a real discussion about which may be the reasons why the CO2 forcing required in this study to improve model-data fit does not fit with proxy-based CO2 estimates.

- Also, methane (as well as gas hydrates) probably have a role to play but given all the issues associated with the absence of real atmospheric dynamics and vegetation feedbacks in the model, I am not sure it can by itself explain the discrepancy between the model and proxy-derived SST, while being forced with reasonable amount of CO2 in the oldest-slices. So widely discussing it do not really help to understand the focus of the paper and the main results. Moreover, as suggested by the title, the discussion should probably focuses on oceanic circulation; however in the present configuration, about half of this section is dedicated to GhGs forcing. Slightly reducing the methane/gas hydrates part, that I think is not very useful in this paper should thus help making the content of the paper reflecting the title

- P15 – I would be helpful to change the title of section 4.2 to something like "Control on benthic temperature".

- P15 – I do not think the title of paragraph 4.3 is meaningful and really relate to its content. From what I am reading it seems that the paragraph mostly relates to caveats of the model framework and uncertainties arising from modeling or unperfect interpretation of data.

- P26 – Paragraph 4.4 is entitled "North Atlantic Salinity and CAS" which is apparently not appropriate as the authors start by discussing orography changes during the Miocene.

It is not clear if the biological pump is represented in the model configuration used for this study (ie. Is the biological pump module activated ?). I think this should be written more clearly, probably in section 2.2, especially because as the authors stated, "the $\delta$13C [...] represents the combined effects of ocean circulation [...] and the ocean biological carbon pump". Anyway, I think that the authors should include a least a short discussion on the eventual changes in the biological pump during the Miocene and how taking it into account could help improving the $\delta$13C model-data fit.

In this study, the orbital parameters are kept as modern. However, at the time the

proxies were generated it is likely that the orbital configuration was different, which can explain also part of discrepancy between proxy and model. I would have liked to see this point being discussed or at least mentioned in the discussion.

Other comments :

- P2 ; l. 43– The rise of mountains belt can also modified the oceanic circulation via changes in atmospheric dynamics (e.g. wind stress) and also via redistribution of atmospheric water fluxes from one basin to the other (e.g. Maffre et al. 2017). It should probably be introduced there as authors refer to such a thing in paragraph 4.4.

- P3 ; l70 – "We employ foraminifera proxy data for: surface ocean temperature, benthic ocean temperature, and benthic ocean $\delta$13C, and compile this for seven time-slices Âż is redundant with l. 85 of the same paragraph "Three different paleo/proxy datasets were compiled: surface temperature, benthic $\delta$18O and benthic $\delta$13C. Âż.

- P3 ; l. 100 " noting that these proxies, like any proxy, are subject to uncertainties and limitations" is a pretty vague statement and do not help the reader to identify potential weakness of the model-data comparison. Short sentence listing mains limitations of that proxies, especially the saturation effect and the bias at high latitudes toward the warm season (e.g. Sluijs et al. 2006 , Bijl et al. 2009 , Richey and Tierney, 2016) would be more useful.

- P7 ; l. 209 – [. . . ]as an additional and independent control on $\delta$13C (but not temperature). Written like that, what you mean is not very clear. Does it mean that you take into account temperature effect on $\delta$13C ?

- P9 ; sub-section 3.1 – The first paragraph on $\delta$13C sounds out of the scope of the section as it is presented from data perspective only, while other paragraphs in this section are exposing data-model comparison. I would therefore suggest that the authors re-organize the section and move this paragraph at the place they discussed the $\delta$13C model-data fit.

- Authors should also be careful to separate CO2 forcing as it is in cGENIE (the one the authors refer to as equivalent CO2 forcing, that take into account other GhGs forcing - for example l. 265 ; P9 and following paragraph) and past CO2 values estimated from proxies (e.g. Sosdian et al. 2018). I found it really confusing because the text switch from one to the other. My advice would be to use "eq. CO2 forcing" when the authors refer to the parameter used as a forcing in cGENIE.

Technical correction

P2 ; l. 34 – Wrong year for the reference – Sepulchre et al. 2014

P2 ; l.55 – Lynch-Stiegltiz, 2003

P5 ; l.134 – "potentially limiting the biological productivity" instead of "potentially limiting to biological productivity"

P10 ; l.292 – "benthic temperature" instead of "benthic T"

P11 ; l. 357 and following, please cite the figure you refer to (Fig. 9) when you state that the model has too cold N. Atlantic compare to data.

l.427 – "greenhouse gases" instead of "green gases"

P.20 ; l.613 – Remove "Bell et al2015"

Fig 2. – Please set the same boundaries for the vertical axis of each plot, otherwise it is very difficult for the reader to read the plot and to understand the differences in data from different time-slices.

Fig 3. – Horizontal scale with the label oriented as they are is difficult to read. A vertical scale, with horizontal labels would probably be more readable.

Fig. 7; Fig. 11 Top - Please keep homogeneous min/max boundaries for the shaded color-scale within the different panels in the same figure. It make it easier to visually compare one time-slice to another. It also highlight changes in the simulated $\delta$13C

pattern, a thing that is not possible when color scale is different within each panel. Fig 7. Add the legend near the shaded color bar (something like "$\delta$13C m-score"), even if it is stated in the caption it helpful for the reader, especially in that case where a lot of information are provided on the same plot.

Fig 7. In the caption, add (from Fig. 5 ....) (from Fig 6...) when components are also shown on another figure. This make is more readable !

Fig. 12 – Please label each panel with alphabet letters, so you can directly refer to the right panel in the text.

References

Bijl et al. 2009, Early Palaeogene temperature evolution of thesouthwest Pacific Ocean, Nature Letters

Richey and Tierney, 2016, GDGT and alkenone flux in the northern Gulf of Mexico: Implications for the TEX86 and UK'37 paleothermometers. Paleoceanography

Butzin et al. 2011, Miocene ocean circulation inferred from marine carbon cycle modeling combined with benthic isotope records. Paleoceanography

Maffre et al. 2017, The influence of orography on modern ocean circulation. Climate Dynamics

Schneider and Schmittner, 2006, Simulating the impact of the Panamanian seaway closure on ocean circulation, marine productivity and nutrient cycling. Earth and Planetary Science Letters

Sepulchre et al. 2014, Consequences of shoaling of the Central American Seaway determined from modeling Nd isotopes. Paleoceanography

Sluijs et al. 2006, Subtropical Arctic Ocean temperatures during the Palaeocene/Eocene thermal maximum. Nature Letters

---

## Referee Comment (RC2) · Anonymous Referee #2 · 17 Feb 2020

The authors use model simulations and proxy records to better understand ocean temperature and circulation change since 15 Ma. To do so, they perform a series of cGENIE simulations using data from previously run HadCM3 simulations with period appropriate topography and 400 ppm $CO_2$. The cGENIE experiments specifically explore the impacts of $CO_2$ concentrations and amounts of freshwater flux from the Atlantic to the Pacific. Equilibrium cGENIE outputs of temperature and d13C are compared with surface temperature and benthic d18O and d13C records. The best model-proxy agreements provide insight into the causes of long term climate change.

This study is interesting. I believe it should be published in Climate of the Past after

revision. The paper is well written and generally well structured. Although there are many limitations of the model setup, these caveats are mostly addressed. Here are some comments that should be addressed before publication -

1) I wonder how important paleogeography is to the response. The discussion contains speculation about the importance of gateways but I think this topic can easily be explored in greater detail. It would be interesting to compare the proxies from all time periods using only the Holocene configurations. Does paleogeography actually improve the model-proxy agreement? The authors already have all the results necessary for this comparison.

2) I would like to see how the biases in HadCM3 are translated into cGENIE (muffin-gen) either by a comparison of the Holocene (HadCM3 forcing) and default cGENIE (observation forcing) simulations or by a comparison of the Holocene ocean circulation in cGENIE and HadCM3. This would help determine if the biases come from HadCM3 or cGENIE. A few sentences on the biases in cGENIE's ocean circulation would also be helpful.

3) In the paper, the FwF changes are usually discussed with respect to changes in North Atlantic deep water. However, do most of the benthic proxy sites record a North Atlantic signal or an Antarctic signal? I realize that the changes are related, but talking about the Antarctic response might be more direct in some instances.

Other comments –

Line 21 – What is the "present-day" climate sensitivity? This is mentioned several times in the text without citation. Is it based on the transient climate sensitivity?

Line 34 – Doesn't Bell et al. (2015) suggest that the closure did not impact AMOC? Please include additional details about ongoing debate within the community here.

Line 50 – Here there are a lot of citations on the importance of vegetation. Are there specific citations to support the roles of bathymetry, topography, and CO2 as well?

Line 85 – List what types of proxies make up the surface reconstructions.

Line 111 – Are the reconstructions on paleolocations.org the same as the reconstructions used to make the maps in the HadCM3 experiments?

Line 133 – Please briefly mention the potential limitations of excluding iron cycling.

Line 165 – How many HadCM3 years are used as inputs?

Line 183 – The argument for a zonal average albedo is not strong because there is no comparison of results with default GENIE simulations.

Line 188 – Why was this CO2 coefficient chosen?

Line 193 – Over how many years are the proxies averaged? I assumed the proxy averaging interval was long enough that orbital variability does not matter much, but based on the supplemental, it seems I was incorrect. I recommend averaging proxy records over 100 kyr or more to reduce the likelihood of capturing high frequency variability that is not simulated with the model. Also, might the use of present-day boreal summer near aphelion skew the results towards particular solutions?

Line 210 – Ha! Hopefully "which does not exist" is not necessary.

Line 219 – "modal"->"model" and "in prep" cannot be cited. Either discuss or remove.

Line 233 – The variability is so small that there is no benefit to averaging? I find this surprising.

Line 239 – I agree this is extremely interesting. Please list the simulations that show this behavior.

Line 243 – For oscillations that are longer (∼4 kyr), how do you know they are persistent with only a 10 kyr long simulation? Please plot an example of these oscillations in the supplement? If you have to average over 4 kyr, are you in equilibrium over these 4 kyr?

What are the initial ocean conditions for these simulations?

Line 266 – Again, what is the present-day climate sensitivity?

Line 269 – Some of this information sounds like a better fit in the methods or figure caption.

Line 277 – Again, you might be able to quantify these things a bit by comparing the proxies against only the Holocene simulation.

Line 299 – Can you modify these assumptions within uncertainty to improve model-proxy agreement?

Line 315 – Not necessary to perform, but I wonder if you might get a better agreement for the Holocene with a lower $CO_2$.

Line 326 – How do you determine the best fit value here?

Line 333 – "again supports"

Line 377 – Interesting!

Line 391 – Given the other parameterizations, how robust is this backed out d13CO2?

Line 413 – Similar result were also found in Carrapa et al., 2019 (Ecological and hydro-climate responses to strengthening of the Hadley circulation in South America during the Late Miocene cooling; PNAS)

Line 427 – "greenhouse"

Line 438 – This paragraph and parts of the following paragraphs do not fit very well with the findings of the simulations and seems a bit unnecessary.

Line 450 – "do"

Line 456 – What about the higher climate sensitivity of the CMIP6 models?

Line 484 – How much flux do you have through the CAS?

Line 487 – Again, you should be able to test the role of the CAS.

[Figure]

Line 495 – The precipitation response to warming is not this simple.

Line 562 – Why is the fit so poor >70°?

Figure 8 – It would be helpful to plot CO2 compilations against your best estimate (e.g. Foster et al., 2017; Berner and Kothavala, 2001).

Figure 11 – Use a single color contour bar.

Figure 14 - Similar cross section plots of d13C and temperature would be very helpful.

---

## Referee Comment (RC3) · Anonymous Referee #3 · 20 Feb 2020

I appreciated this paper in the spirit of "all models are wrong, but some are useful". I don't mean this to trivialize the work here, but to note that a number of very important constraints to the model, such as deep ocean temperature evolution, are poorly known. That said, a real benefit to the approach in comparison to other efforts to model the middle Miocene is that CO2 is not treated as a known, but as a primary variable to be estimated from the model. Also, the model calculates a salinity flux adjustment to try to capture basin scale patterns- this should make an interesting target for data generators. Overall, what I appreciated about the approach here is that it takes multiple paleo targets into consideration, rather than simply trying to capture, for example, average global temperatures or global surface temperature gradients. The interplay the model

allows between different types of paleo observations is very useful to a data generator like me.

The authors readily acknowledge that their CO2 estimates are better understood as "CO2-equivalent" rather than as direct CO2 levels. Some in the community may want to hammer the high CO2 estimates as completely unrealistic. However, I align with those who don't think the available CO2 estimates are very reliable and I urge the authors not to back off from their estimates as CO2 too readily (admitting that vegetation and methane likely do contribute significant warming to the real Miocene world).

Along the way, the authors try to point out which kinds of inferences are strongly constrained by different data types and which are not. This is very useful to someone not using cGENIE- it helps them interpret what's robust and what's not. But the presentation here often just says "X constrains Y much more than Z does" without taking the step of diagnosing how the authors know that. A little more explanation would let the non-modeler get much more out of this analysis.

Because of the breadth of this study, I necessarily have questions that I'd like to see clarified:

Model set up: • -can the authors assess the effect the lack of interactive clouds has on the sensitivity of surface temperatures to CO2? • – Does the model allow for Albedo- vegetation feedbacks (I assume not)? As above, does this mean it will tend to underestimate Earth System Sensitivity to CO2? • GCM input generated with CO2 @ 400 ppmv • • Lines 175-182 (geographic/bathymetric choices such as not including Med, Greenland-NA connection, Bering Sea open/closed): can the authors provide the reader with a sense of whether these affect model results significantly • Specify model sensitivity to CO2? The text specifies "which is as the present day" without giving a value?

Emphasis on bathymetry and closing/opening of seaways on deep ocean properties: • is it justified in comparison to more subtle modulators such as sea ice, ridge

bathymetry, small changes in T and S? I'm thinking of ideas such as put forth for Ferrari et al. to account for glacial storage of CO2 that rely on mixing or lack thereof of the North Atlantic and Southern Ocean deep circulation loops ⇢ I'm not sure why the authors have an open CAS for all except the Holocene time slice, since they note how circulation between the deep North Atlantic and Pacific affect model fits of carbon isotope gradients and North Atlantic salinity ⇢ Circulation and deep ocean CO2 inventory- is this examined explicitly? We know from Pleistocene CO2 cycles that the storage of carbon in the Southern Ocean can give a 80-90 ppm CO2 effect. Some theories (e.g. R. Ferrari et al. 2014 ) hypothesize that the connection between the North Atlantic and Southern ocean loops is critical to whether CO2 is stored in the deep ocean or vented to the atmosphere. Can the authors comment on how the Southern Ocean CO2 pump works in cGENIE and if the model is capable of capturing changes such as proposed by Ferrari on carbon storage?

Model-data Methodology ⇢ Line 70 : We employ foraminifera proxy data for: surface ocean temperature, Vs Lines 100-101 Published surface temperature data selected are those using either alkenones or TEX86 for all seven slices (Figure 1 for surface 100 temperature data locations and Supplementary material ⇢ A further question: what is the balance of alkenone vs TEX86 data? Does the balance change between time slices? Which TEX86 index was used?

Benthic isotopes & temperature (105-107)?? ⇢ It's hard to follow this logic: "These species were selected so that temperature could be calculated from $\delta$18O using 105 Marchitto et al. (2014). Final benthic temperatures calculated from $\delta$18O take account of the effect of benthic water salinity on $\delta$18O which is affected by ocean circulation (the temperatures in Table S2 are uncorrected for salinity)." As the deep ocean has extremely small salinity variations, it's not clear why a salinity correction is needed. The deep ocean salinity will tend to be pretty homogeneous, because it is filled from only a handful of sources. Furthermore, the deep ocean salinity is constrained to be very close to the average salinity of the entire ocean, since only the thin skin layer of the

ocean exchanges water with the atmosphere. The only way to make large changes in average salinity on the timescale considered here is to remove/add water by forming/melting ice. The correction for the ice volume effect on isotopic composition, is potentially large and not very well constrained from the Pliocene and earlier, as the volume and the isotopic composition of polar ice aren't well known. (see e.g. Bohaty et al 2012, EPSL for an example taken from the Eocene/Oligocene transition). • In line with the latter point, how large do the authors think the deep ocean temperature uncertainties are, and how do they evaluate this? • Clarify this: "Finally, local salinity has a strong control on $\delta$18O; an increase in measured benthic $\delta$18O in the N. Atlantic during the late Miocene that may be interpreted as evidence of a strong cooling, may actually be attributable to the increased salinity of the deep sea water, where salinity (rather than temperature) dominates the $\delta$18O signal recorded in the benthic foraminifera. With the onset of Atlantic overturning circulation during the Miocene, the salinity of deep N. Atlantic waters has a strong control on $\delta$18O, and when 470 included in the temperature calculation results in increases of up to 3°C in some locations (compared to the temperature uncorrected for salinity) (Fig. 10)." The salinity of the global deep ocean is constrained to be the average salinity of the ocean (because deep water masses have only one or two sources and must have nearly the same densities throughout the global deep ocean); likewise for the d18O value of the deep ocean water. Therefore, I think the language above should be clarified so the "the d18O signal recorded" is rewritten as "the d18O signal recorded in the North Atlantic. . ." or similar. I take it from the topic sentence of the paragraph that the authors are looking at local patterns but the reader may mistake the intent of the later sentences.

Carbon isotopes • d13C data: don't capture changes in terrestrial carbon storage. I was curious if it could affect the predicted atmospheric d13C (Figure 13, lines 387-392). The authors might also note that a change in atmospheric d13C would affect the average d13C of photosynthate. I'd like to see more discussion of how the atmospheric d13C was derived- the presentation here is too short to be useful to someone not intricately familiar with cGENIE. • Although it's an older paper, this one was a nice

one that I didn't see cited: Woodruff, F. and S. M. Savin (1989). "Miocene deepwater oceanography." Paleoceanography 4(1): 87-140.

• D13C as a tracer of circulation: I wondered if the d13C data base has a change in the proportion of Uvigerina vs Cibicidoides values over time? While we "correct" for isotope offsets between species, this may not always work.

Finally, the style of the paper wavers a bit between assuming no previous background and a lot of background. I'm think this level of explanation is not needed for the target reader- it's too low-level (lines 89-92): "The isotope of carbon, carbon-13, has been used as a tracer for paleo ocean circulation for many years (Lynch-Stieglitz 2003). It is a stable isotope, heavier than carbon-12 and accounts for about 1% of all carbon on Earth. The ratio of carbon-13 90 to carbon-12, designated as "$\delta$13C" – the divergence from a standard in units of parts-per-thousand (‰, can be estimated for paleo ocean waters by measuring the $\delta$13C in shells of foraminifera formed in those paleo water masses. . .. Shells of dead foraminifera gradually accumulate on the ocean floor, thus providing a record of changes in water column chemistry over time at that location in the form of ocean sediments"

---

## Author Comment (AC1) · 5 Aug 2020

Response to comments from reviewers

"Data-constrained assessment of ocean circulation changes since the middle Miocene in an Earth system model" by K.A.Crichton et al

An Overview of changes we propose to the manuscript based on the reviewers comments:

- Add simulations of a closed CAS from 10Ma and determine whether the model can better reproduce the data (as we have suggested in the manuscript). (Indeed, we have already created alternative continental configurations and re-run the pCO2 vs. FwF ensembles for each of these.) We have already run statistics for benthic $\delta^{13}C$ for a closed CAS between 10Ma to present, and model skill scores are higher than for the open CAS cases. We will perform the full analysis for temperature and $\delta^{13}C$ on this new ensemble as for the one in the original manuscript. We will add to our discussion using the results of this new model ensemble.
- Better illustrate the CAS configurations that we apply.
- Add a specific discussion on cGENIEs fixed climate sensitivity and its implication for pCO2 levels.
- Add ocean cross section plots for temperature and $\delta^{13}C$ distribution as figure 14.
- Add a model sensitivity test of orbital configuration in supplementary material.
- Add to the supplementary material a detailed comparison of the 'standard' cGENIE climatology and ocean circulation vs. cGENIE configured following the late Holocene ('modern') HadCM3 simulation.
- Add some more information on sustained oscillations to supplementary material.
- Add a brief overview of ocean carbon storage changes in our ensembles in the supplementary material.
- Make improvements and revisions throughout the main text as recommended by reviewers (itemised below in specific response to their comments).

**Reviewer 1**

CAS configuration:

*"As most of the discussion is CAS-oriented, authors should probably provide a few lines about CAS configuration changes during the Miocene and effect on the global circulation in the introduction using existing literature"*

We will add more background from previous studies on CAS and its impact on circulation and climate in the introduction. We can and will also now link this discussion to the new alternative model continental configurations of CAS open vs. closed.

*"As authors refers a lot to CAS closure, they should also probably provide as supplementary information a table that contains at least the mean depth of CAS in the different settings they used."*

We will add information on the depth and width of the model CAS in the supplementary material. Specifically – although Figure 3 shows the global bathymetry of each of the model continental configurations, we will add regional blow-up maps to better show the CAS bathymetry in model

detail. To this, we will add surface and sub-surface ocean velocity maps to illustrate the model simulated circulation through the CAS (and contrast with previous GCM-based modelling studies of this time interval).

*"The paper would obviously benefit from additional sensitivity tests on CAS closure, especially for younger time slices (e.g. 2.5 and 4.5 Ma), as it seems, from what the authors states, that this is the missing point to be able to improve model-data fit."*

We will carry out and present results from further simulations with a closed CAS (from 10Ma) to support our conclusions regarding CAS closure and $\delta^{13}C$ benthic patterns (and indeed have already completed the cGENIE ensemble simulations and are processing the model vs. data statistics). We will additionally include an analysis of the modern continental configuration, but with an open CAS, as a test of the data constraints in distinguishing an open vs. closed CAS.

Discussion:

*"Moreover, as suggested by the title, the discussion should probably focuses on oceanic circulation; however in the present configuration, about half of this section is dedicated to GhGs forcing."*

Rather than change the title, we will extend the discussion about ocean circulation, particularly in light of the new model ensembles and analysis with CAS open and the additional of new analysis of circulation flow through the CAS. We will also make it clearer at the start of the manuscript why we vary GHG forcing (as well as fresh water flux adjustments) and how this relates to constraining ocean circulation.

*"As the section 4.1 is written now, it looks more like a list of previous Miocene studies than a real discussion about which may be the reasons why the CO2 forcing required in this study to improve model-data fit does not fit with proxy-based CO2 estimates."*

In revising the manuscript, we will pay particular attention to this section. We will provide a new figure contrasting model-generated pCO2 estimates with currently available proxies (as requested elsewhere), but as per the clarifications we will provide regarding 'why vary GHGs? if the study is focussing on ocean circulation' (the point above) we will make clear that this is a study designed to constrain surface climate not GHGs. Varying radiative forcing in the model is a means-to-an-end in firstly modulating the large-scale circulation of the ocean and hence benthic data patterns, but also in adjusting the mean benthic temperature and hence enabling us to employ more involved model-data stats such as the Model Skill Score rather than just e.g. the correlation coefficient.

*"Also, methane (as well as gas hydrates) probably have a role to play but given all the issues associated with the absence of real atmospheric dynamics and vegetation feedbacks in the model, I am not sure it can by itself explain the discrepancy between the model and proxy-derived SST, while being forced with reasonable amount of CO2 in the oldest-slices. "*

We do not propose that all the discrepancy is due to a lack of methane in the model (and acknowledge the limitations of our model that has no land surface representation). We will clarify this in the text. However, methane is a possible source of any discrepancy between our equivalent $CO_2$ and the actual Miocene $CO_2$ (which is not necessarily the proxy-data indicated value), so we consider it an important factor. We will streamline the text on methane to ensure its role is not too over-emphasised with respect to other sources of model eq.$CO_2$-real$CO_2$ mismatch (e.g. vegetation interactions etc.). We will also add new discussion about (assumed model) climate sensitivity and

the implications for model-required pCO2 values, as in fact the proxy temperature constraints are arguably on radiative forcing in the model rather than pCO2 per se.

*"It is not clear if the biological pump is represented in the model configuration used for this study (ie. Is the biological pump module activated ?). I think this should be written more clearly"*

We will clarify this in the text by adding a paragraph to summarise the model set up for ocean carbon cycling.

*"In this study, the orbital parameters are kept as modern. However, at the time the proxies were generated it is likely that the orbital configuration was different, which can explain also part of discrepancy between proxy and model. I would have liked to see this point being discussed or at least mentioned in the discussion."*

We will provide the results of a model sensitivity test of changing the orbital configuration. We did take account of orbital configuration for the selection of benthic data, by aiming for a mean (or representative) value in a window of +-1 million years. We will add this to the text in the discussion, as well as in discussion of the surface temperature data, in terms of orbital configuration, that we obtained from other studies.

*"Authors should also be careful to separate CO2 forcing as it is in cGENIE (the one the authors refer to as equivalent CO2 forcing... My advice would be to use "eq. CO2 forcing" when the authors refer to the parameter used as a forcing in cGENIE."*

In the text we will ensure that whenever we talk about the model forcing we will use $eq.CO_2$ to avoid confusion.

Other comments
We will apply all suggestions made in this section.

Technical corrections
We will apply all technical corrections except this one:

*"Fig. 7; Fig. 11 Top - Please keep homogeneous min/max boundaries for the shaded color-scale within the different panels in the same figure. It make it easier to visually compare one time-slice to another. It also highlight changes in the simulated 13C"*

We will apply the homogenous colour scale for figure 7, but not for figure 11. The $\delta^{13}C$ global value changes during the study period, by ~1‰ according to our figure 13 for the atmosphere (well mixed) value. Keeping the colour bar the same for all timeslices in this figure 11 would mean any signal (from atmosphere – ocean differences) would be entirely dominated by this $\delta^{13}C$ global value. The current colour scale in figure 11 shows the relationship between the atmosphere and the benthic $\delta^{13}C$ value, where in some timeslices the benthic value is closer or further away from the atmosphere (or surface ocean value). We will clarify this in the text.

**Reviewer 2**

*"1) I wonder how important paleogeography is to the response. The discussion contains speculation about the importance of gateways but I think this topic can easily be explored in greater detail. It would be interesting to compare the proxies from all time periods using only the Holocene configurations. Does paleogeography actually improve the model-proxy agreement? The authors already have all the results necessary for this comparison."*

As per in reply to Reviewer 1 -- we will (and indeed have done now) perform CAS closed simulations for timeslices from 10Ma to test our original conclusion (that $\delta^{13}$C data can be better explained with a closed CAS from 10Ma). To the specific point of performing model-data comparison using only the Holocene configuration – instead we propose the other way around – using the new open CAS Holocene configuration we have also generated, and contrasting with the Holocene data, the point being to test whether the most complete (Holocene) data set can distinguish open from closed CAS (when we know the answer for modern is 'closed'). This point is additionally addressed via the new closed CAS 2.5 through 10.0 Ma simulations.

*"2) I would like to see how the biases in HadCM3 are translated into cGENIE (muffingen) either by a comparison of the Holocene (HadCM3 forcing) and default cGENIE (observation forcing) simulations or by a comparison of the Holocene ocean circulation in cGENIE and HadCM3. This would help determine if the biases come from HadCM3 or cGENIE. A few sentences on the biases in cGENIE's ocean circulation would also be helpful."*

We will include in the supplementary material a formal comparison of the HadCM3 forced vs the original observations-forced configuration of cGENIE, including an analysis of ocean circulation patterns and climatology.

*"3) In the paper, the FwF changes are usually discussed with respect to changes in North Atlantic deep water. However, do most of the benthic proxy sites record a North Atlantic signal or an Antarctic signal? I realize that the changes are related, but talking about the Antarctic response might be more direct in some instances."*

We will include discussion of any possible Antarctic signal with respect to the benthic data in the text.

Other comments

We will implement all comments suggested here. Some further explanation for selected comment is as follows:

*"Line 111 – Are the reconstructions on paleolocations.org the same as the reconstructions used to make the maps in the HadCM3 experiments?"*

They are not the same but we will provide error estimates in supplementary material between the two reconstructions (the differences are fairly small, we have now done this comparison). We found an error in one paleolocation in the datatables (site 1120), we will correct this in a revised version will re-run the statistical analysis on the full ensemble, but do not expect any significant change in the results of the statistical analyses (and certainly no change in our main conclusions).

*"Line 188 – Why was this CO2 coefficient chosen?"*

This is the value predicted in the original carbon cycle tuning of Ridgwell et al. [2007] using the scaling from Wanninkhof [1992] in conjunction with climatological winds. In moving to a GCM-derived wind field, the spatial and temporal averaging going into the wind product is very different, meaning that one would expect the scaling between the wind speed product and gas exchange rate to change. We determine this new (GCM-based) tuning on the basis of retaining approximately the same global annual mean gas transfer coefficient for $CO_2$ as before. We will make this much clearer in the revision, and likely expand on this as part of the new discussion regarding climatology vs. GCM-derived cGENIE configurations (addressing previous points).

*"Line 193 - Over how many years are the proxies averaged? I assumed the proxy averaging interval was long enough that orbital variability does not matter much, but based on the supplemental, it seems I was incorrect. I recommend averaging proxy records over 100 kyr or more to reduce the likelihood of capturing high frequency variability that is not simulated with the model. Also, might the use of present-day boreal summer near aphelion skew the results towards particular solutions?"*

The data points were selected to be broadly representative of the pattern in the $\delta^{18}O$ or $\delta^{13}C$ over a window of +-1million years. This was not done by averaging over a certain interval but by visual inspection of the timeseries about the target age. The value selected was the mid-range of any evident glacial-interglacial cycling. Some of the data has far lower resolution than that would permit this method, so a window of +-1 million years was inspected to determine the "representiveness" of a data value. This approach was taken due to the large uncertainty in age models from sediment core data, and in many datasets being far lower resolution than orbital cycles scales (e.g. less than one datapoint every 50,000 years for example). So in essence, we have taken account of orbital variability (where we can). To address the question of orbital configuration, we will add a model sensitivity test of changing orbital configuration to the supplementary material.

*"Line 233 – The variability is so small that there is no benefit to averaging? I find this surprising."*

The simplified 2D nature of the atmospheric component, in lacking interactive winds, leads to no interannual variability being present in the model. Hence at steady state, every year is effectively 'the same'. We will clarify this in the main text.

*"Line 239 – I agree this is extremely interesting. Please list the simulations that show this behavior."*

We are happy to add an illustration of this information to supplementary material.

*"Line 243 – For oscillations that are longer (~4 kyr), how do you know they are persistent with only a 10 kyr long simulation? Please plot an example of these oscillations in the supplement? If you have to average over 4 kyr, are you in equilibrium over these 4 kyr?"*

We see how our original text is a little misleading, there are actually no oscillations in any simulation longer than about 4.5 kyr. We will clarify the specific range we have determined. Will also provide an example of a 50 kyr long simulation in SI to illustrate the long-term persistence and stability of oscillations (also further addressing the previous request for more information).

*"Line 299 – Can you modify these assumptions within uncertainty to improve model-proxy agreement?"*

We will add additional short explanation as to the impact of these assumptions in the text, where it may have an impact on global ice volume at 15Ma. However, a more extensive analysis of the impact of the uncertainty in global ice volume is probably a step too far. Temperature scales linearly with

changes in global ice volume (in the paleotemperature calculation that we apply), and as we don't have a strong error estimate for the global ice volume it is difficult to determine the impact on our results (except to say as we have that it has an impact in a certain direction).

*"Line 326 – How do you determine the best fit value here?"*

We only select values that we have tested (so, no intermediate values between 280ppm and 400ppm). A certain amount of the best-fit selection is a question of judgment, but we have tried to defend our selections in the text. In this case, 280ppm or 400ppm can be considered a good fit, but comparing to the Holocene range, a higher CO2 (than Holocene) seems to be suggested. In figure 8 we note these high uncertainties by plotting a range of values (shaded grey) that also show some agreement with our data constraints.

*"Line 487 – Again, you should be able to test the role of the CAS."*

We will include the results of just such a test using our new series of alternative closed CAS simulation.

*"Line 562 – Why is the fit so poor >70°?"*

The high southern latitudes are far more effected by the polar front in the value of water $\delta^{18}O$, where sea-ice plays a strong role. Other latitudes follow the more simple salinity relationship. We will add some explanation to the text.

*"Figure 8 – It would be helpful to plot CO2 compilations against your best estimate (e.g. Foster et al., 2017; Berner and Kothavala, 2001)."*

The CO2 forcing that we find is an equivalent CO2, so plotting it together with data-estimated CO2 values is probably misleading. However, we can add a Figure of recent pCO2 estimates, but in addition to plotting the cGENIE model-required pCO2 values, we will also plot (i) model- required pCO2 values corrected for potentially elevated atmospheric pCH4, and (ii) model- required pCO2 values transformed using a different climate sensitivity assumption – i.e. what the model and temperature data actually constrains is a radiative forcing, which in cGENIE equates to a particular pCO2 value via the assumed climate sensitivity, hence assuming e.g. a higher climate sensitivity value as per recent fully coupled GCM results, would result in a lower required pCO2 estimate for cGENIE. Note that the fixed/imposed and non state-dependent climate sensitivity in cGENIE allows us to simply re-calculate pCO2 and not re-run the model ensembles.

*"Figure 11 – Use a single color contour bar.*
*Figure 14 - Similar cross section plots of d13C and temperature would be very helpful."*

As with reviewer 1 comment, we will not change the colour bars for figure 11. We will clarify in the text the value of keeping the colour bars on this sliding scale. We can add cross sections plots for temperature and d13C and will keep all colour bars on the same scale, which will address the point about comparing d13C absolute values.

**Reviewer 3**

*"But the presentation here often just says "X constrains Y much more than Z does" without taking the step of diagnosing how the authors know that. A little more explanation would let the non-modeler get much more out of this analysis."*

We will go through the text to identify where this occurs and improve the explanations.

Model set up:

*"- can the authors assess the effect the lack of interactive clouds has on the sensitivity of surface temperatures to CO2?"*

As per our reply to Reviewer 2 and questions about the model-required pCO2 values, we will add some discussion about the (fixed) sensitivity in cGENIE and the implications for the diagnosed pCO2 values, including e.g. the absence of cloud feedbacks in the model that might otherwise lead to state-dependence of climate sensitivity.

*"— Does the model allow for Albedo- vegetation feedbacks (I assume not)? As above, does this mean it will tend to underestimate Earth System Sensitivity to CO2?"*

According to some studies (e.g. Bradshaw et al 2015) yes, without vegetation feedbacks the model may underestimate Earth system sensitivity to CO2. We discuss this in the discussion section, but we will clarify that by adding "we discuss climate sensitivity of cGENIE and the implications for our results in the discussion" in the model description.

*"- nGCM input generated with CO2 @ 400 ppmv"*

We are happy to clarify and add discussion to the text in respect to the use of a series of GCM simulation assuming a single specific pCO2 value to provide climatological surface boundary conditions, whilst significantly varying pCO2 in the cGENIE model ensembles. This is related again to the fixed/imposed climate sensitivity in cGENIE, where the pCO2 levels should be seen as a radiative forcing, this point from reviewer 3 can be addressed by discussion (and new figures) of the impact of methane and different climate sensitivity (as in response to reviewer 2) on the pCO2 imposed.

*"Lines 175-182 geographic/bathymetric choices such as not including Med, Greenland-NA connection, Bering Sea open/closed: can the authors provide the reader with a sense of whether these affect model results significantly"*

Previous studies have suggested that these particular sea-ways are less important to global circulation changes in the Miocene-to-present than the Drake Passage and CAS. As such, we do not quantify specifically the effect of these seaways on the result of our analysis except to assume they are less important (based on other studies) than the Drake Passage and Central American Seaway. We will add a simulation of a closed CAS from 10Ma, so the CAS is in fact the only seaway we can analyse for its effect on circulation (after that closed CAS simulation is complete). Currently, we can compare the model output at different timeslices, but for the same forcings, to have an overall view of the effect of paleogeography (e.g. figure 5) but we can't attribute specific amounts of this effect to a particular seaway (this is outside the scope of this study).

We will explore much more clearly CAS bathymetry and bathymetry differences, and the consequences of this with the benefit of the new closed CAS ensemble.

*"- Specify model sensitivity to CO2? The text specifies "which is as the present day" without giving a value?"*

It is approximate 3°C for a doubling of pCO2. We will add this value, and state in the model description that we will further discuss climate sensitivity in the discussion.

*"Emphasis on bathymetry and closing/opening of seaways on deep ocean properties: is it justified in comparison to more subtle modulators such as sea ice, ridge bathymetry, small changes in T and S? I'm thinking of ideas such as put forth for Ferrari et al. to account for glacial storage of CO2 that rely on mixing or lack thereof of the North Atlantic and Southern Ocean deep circulation loops"*

There is large uncertainty as to the CO2 forcing and circulation states (and paleogeography) in this time period and we consider these to be the first order controllers on the global climate for the purposes of this study. Glacial storage of carbon is also an area of study in which uncertainties exist, but large scale paleogeography and CO2 levels were (essentially) knowns for that time period (unlike ours).

We are aiming to create plausible "mid-range" (i.e. not extreme glacial or peak interglacial) climate and circulation states for our timeslices since the middle Miocene. The model determines sea-ice extent internally. In this sense, sea-ice is a result of climate state (not the inverse) for our study, but of course interacts with ocean physics and the carbon cycle. Ridge bathymetry can be seen as a smaller scale effect than the changes in bathymetry we consider that result in e.g. open or closed seaways for the Miocene period. Small changes in T and S are determined by our model physics: T we control with CO2 forcing and S is (to an extent) controlled by our flux correction variable for the N.Atlantic. We consider that CO2 forcing and circulation driven by that forcing and by paleogeography are much more significant controllers on global climate than those listed by the reviewer.

What we can easily do to aid here, is to add (to SI) an analysis of carbon storage in the ocean, which then for a single assumed value of pCO2, provides the explicit information regarding the impact of changing circulation on stored carbon, and between ensembles provides information about how continental configuration (and circulation) impacts stored carbon. We note that although cGENIE incorporates a variety of e.g. preformed tracers, they were not included in the model configuration used in the ensembles. Adding then and then re-running and analysing the ensembles would admittedly be super interesting, but now way outside the scope of the present paper.

*"I'm not sure why the authors have an open CAS for all except the Holocene time slice, since they note how circulation between the deep North Atlantic and Pacific affect model fits of carbon isotope gradients and North Atlantic salinity"*

The CAS is open for all but the Holocene timeslice due to the method we apply for regridding the HadCM3 model, which is of finer resolution than cGENIE. We will add a new set of simulations with a closed CAS from 10Ma to test our conclusions (see replies to previous points) and also make clearer how the first set of continental configurations came to be (in terms of the re-gridding algorithm).

*"Circulation and deep ocean CO2 inventory- is this examined explicitly? We know from Pleistocene CO2 cycles that the storage of carbon in the Southern Ocean can give a 80-90 ppm CO2 effect. Some theories (e.g. R. Ferrari et al. 2014 ) hypothesize that the connection between the North Atlantic and Southern ocean loops is critical to whether CO2 is stored in the deep ocean or vented to the*

*atmosphere. Can the authors comment on how the Southern Ocean CO2 pump works in cGENIE and if the model is capable of capturing changes such as proposed by Ferrari on carbon storage?"*

See reply above – yes we can add a brief analysis of ocean carbon storage and how continental configuration and circulation patterns impact this. Associated with this we can also include a very brief comparison with ocean LGM carbon storage, but limited to avoid straying too far into off-topic glacial/interglacial questions.

*"what is the balance of alkenone vs TEX86 data? Does the balance change between time slices? Which TEX86 index was used?"*

We will add a note stating which referenced surface temperature set is from which proxy in the datatable.

*"As the deep ocean has extremely small salinity variations, it's not clear why a salinity correction is needed. The deep ocean salinity will tend to be pretty homogeneous, because it is filled from only a handful of sources. Furthermore, the deep ocean salinity is constrained to be very close to the average salinity of the entire ocean, since only the thin skin layer of the ocean exchanges water with the atmosphere. The only way to make large changes in average salinity on the timescale considered here is to remove/add water by forming/ melting ice. The correction for the ice volume effect on isotopic composition, is potentially large and not very well constrained from the Pliocene and earlier, as the volume and the isotopic composition of polar ice aren't well known… The salinity of the global deep ocean is constrained to be the average salinity of the ocean (because deep water masses have only one or two sources and must have nearly the same densities throughout the global deep ocean); likewise for the d18O value of the deep ocean water. Therefore, I think the language above should be clarified so the "the d18O signal recorded" is rewritten as "the d18O signal recorded in the North Atlantic: : :" or similar. I take it from the topic sentence of the paragraph that the authors are looking at local patterns but the reader may mistake the intent of the later sentences."*

Supplementary material figure A2 of deep ocean $\delta^{18}O_{sw}$ demonstrates the need to correct for $\delta^{18}O_{sw}$ spatial variability in the paleotemperature calculation (rather than assuming a global fixed value). Supplementary figure A1 demonstrates the effect of this spatially variable $\delta^{18}O_{sw}$ on the calculated temperature. Salinity strongly affects $\delta^{18}O_{sw}$, and this salinity value comes not only from global ice volume (where ice is fresh, so the ocean is more saline for large ice volume) but from everyday effects of evaporation and precipitation at the ocean surface, and from these water masses circulations, which transport more or less saline water to the deep; the "global ocean conveyer" is the thermohaline circulation (haline referring to salinity).

The cGENIE model traces salinity through evaporation, precipitation and circulation, so we use cGENIE salinity to correct for local $\delta^{18}O_{sw}$. As salinity affects $\delta^{18}O_{sw}$ (again, due to evaporation and precipitation because less energy is required to evaporate $^{16}O$ than $^{18}O$, and less energy is required to precipitate $^{18}O$ than $^{16}O$) and these more or less salty waters are differently distributed (by ocean circulation), the deep ocean is not homogenous in $\delta^{18}O_{sw}$ and we need to correct for this non-homogenous water $\delta^{18}O_{sw}$ *at every location* to get temperature from foram shells $\delta^{18}O$ (using the Marchitto et al 2014 paleotemperature calculation). As we think the thermohaline circulation with the Atlantic overturning leg established in its present form since the middle Miocene, this correction for local $\delta^{18}O_{sw}$ becomes even more important for our study. By applying this salinity correction to $\delta^{18}O_{sw}$ some benthic temperature estimates change by up to 3°C, which is clearly a significant amount. The correction (and need for it) is described in Appendix A. We will go through the text to ensure it is well sign posted and clear.

*"I'd like to see more discussion of how the atmospheric d13C was derived- the presentation here is too short to be useful to someone not intricately familiar with cGENIE."*

The ocean $\delta^{13}$C scales linearly with atmospheric $\delta^{13}CO_2$. We simply used the difference between the data mean benthic d13C values and the mean model benthic $\delta^{13}$C values (which were all forced with d13CO2 of -6.5) and used the offset between these to determine the $\delta^{13}CO_2$ – i.e. the $\delta^{13}CO_2$ value that would minimise the mean model-data benthic $\delta^{13}$C offset. We will add a clarifying sentence to the existing text.

*"Woodruff, F. and S. M. Savin (1989). "Miocene deepwater oceanography." Paleoceanography 4(1): 87-140."*

We use data from this study in our dataset. In general, we did not want to rely on existing data-interpretation studies to interpret our model-data results, as we rely on the globally distributed datasets from many sources to constrain circulation (rather than specific early studies).

*"I wondered if the d13C data base has a change in the proportion of Uvigerina vs Cibicidoides values over time? While we "correct" for isotope offsets between species, this may not always work."*

We only use Planulina and Cibicidoides species to compile our dataset in order that we can apply the specific paleotemperature calculation of Marchitto et al 2014 (this is already specified in the text). We are aware that different species have different sensitivities to temperature for their shells $\delta^{18}$O, hence why we restrict the species.

*"Finally, the style of the paper wavers a bit between assuming no previous background and a lot of background. I'm think this level of explanation is not needed for the target reader- it's too low-level (lines 89-92): "The isotope of carbon, carbon-13, has been used as a tracer for paleo ocean circulation for many years (Lynch-Stieglitz 2003). It is a stable isotope, heavier than carbon-12 and accounts for about 1% of all carbon on Earth. The ratio of carbon-13 90 to carbon-12, designated as " 13C" – the divergence from a standard in units of parts-per-thousand (‰, can be estimated for paleo ocean waters by measuring the 13C in shells of foraminifera formed in those paleo water masses: : :. Shells of dead foraminifera gradually accumulate on the ocean floor, thus providing a record of changes in water column chemistry over time at that location in the form of ocean sediments""*

We will go through the manuscript to ensure a more consistent level of detail and assumed reader background.

We thank all reviewers for their comments.

---

## Author Response (AR2)

Authors response to corrections for cp-2019-151

"Data-constrained assessment of ocean circulation changes since the middle Miocene in an Earth system model" Crichton et al.

All reviewers comments and technical corrections were implemented, and marked up in the corrected version, including:

- Some changes to introductory paragraph on mid-Miocene CO2 levels
- Adding citations for more appropriate references for the disappearance of the Tethys Sea
- Reviewer requested R2 values for figures 9 -11, instead we point readers to our model-data fit statistic M-score in figure 5

Other very small changes to the text requested by co-authors – and is marked up in the corrected version, and do not change any of the meaning of the content of the accepted version.